# *Mylk3* null C57BL/6N mice develop cardiomyopathy, whereas *Nnt* null C57BL/6J mice do not

Jack L Williams[1], Anju Paudyal[2], Sherine Awad[1], James Nicholson[1], Dominika Grzesik[1], Joaquin Botta[1], Eirini Meimaridou[3], Avinaash V Maharaj[1], Michelle Stewart[2], Andrew Tinker[4], Roger D Cox[5], Lou A Metherell[1]

The C57BL/6J and C57BL/6N mice have well-documented phenotypic and genotypic differences, including the infamous nicotinamide nucleotide transhydrogenase (*Nnt*) null mutation in the C57BL/6J substrain, which has been linked to cardiovascular traits in mice and cardiomyopathy in humans. To assess whether *Nnt* loss alone causes a cardiovascular phenotype, we investigated the C57BL/6N, C57BL/6J mice and a C57BL/6J-BAC transgenic rescuing NNT expression, at 3, 12, and 18 mo. We identified a modest dilated cardiomyopathy in the C57BL/6N mice, absent in the two B6J substrains. Immunofluorescent staining of cardiomyocytes revealed eccentric hypertrophy in these mice, with defects in sarcomere organisation. RNAseq analysis identified differential expression of a number of cardiac remodelling genes commonly associated with cardiac disease segregating with the phenotype. Variant calling from RNAseq data identified a myosin light chain kinase 3 (*Mylk3*) mutation in C57BL/6N mice, which abolishes MYLK3 protein expression. These results indicate the C57BL/6J *Nnt*-null mice do not develop cardiomyopathy; however, we identified a null mutation in *Mylk3* as a credible cause of the cardiomyopathy phenotype in the C57BL/6N.

## Introduction

Differences in background genetics can produce many different baseline phenotypes in the ever-growing library of laboratory mouse strains, and these can have profound influences on the expression of particular trait(s) (1, 2, 3, 4, 5). In the last decade, a body of research has begun to enumerate these differences, including differences in metabolism, disease susceptibility, cardiovascular function, behaviour, and many more (6, 7, 8, 9). Particularly well studied are the C57BL/6 substrains; the C57BL/6J (B6J), previously the most widely used substrain, and the C57BL/6N (B6N), which the International Mouse Phenotyping Consortium (IMPC) has chosen as the background for their 20,000 gene knockouts (https://www.mousephenotype.org/). Many genetic differences are known to exist between these two closely related substrains with perhaps the most commonly studied the nicotinamide nucleotide transhydrogenase (*Nnt*) mutations in the B6J.

In 2012, *NNT* was identified as the causative gene in 15 families with familial glucocorticoid deficiency, a disorder of adrenal insufficiency without heart involvement (10, 11). Subsequently, a further 19 families have been reported, with a single, highly consanguineous patient developing hypertrophic CM with a homozygous *NNT* mutation and three monoallelic variants have been linked to left ventricular non-compaction (10). A number of publications have linked NNT expression levels to cardiovascular traits, including remodelling differences, hypertension, and heart growth in mice and zebra fish (12, 13, 14, 15, 16). Surprisingly, research from the Maack Lab showed the NNT-null B6J mouse is protected from oxidative stress and heart failure in comparison with the B6N mouse, possibly because of a "reverse activity" of NNT in the B6N mouse (14). Furthermore, the NNT-replete B6N mice show increased fibrosis in response to chronic angiotensin II treatment, along with increased expression of cardiac remodelling genes; however, the authors demonstrate these changes are not due to NNT status (15). B6N mice are more susceptible to transaortic constriction than B6J with B6N having lower survival rates and cardiac function (17). The paucity of epidemiological evidence combined with the conflicting data in mouse models means NNT's contribution to cardiac phenotypes is uncertain.

In this study, we aimed to define the role of NNT in CM using the B6J NNT-null mice and a rescue model in the B6J mice, restoring NNT expression by expression of a bacterial artificial chromosome with the full-length *Nnt* gene (C57BL/6J-BAC, NNT transgenic [B6J-Nnt]) (18). We also compared both of these strains with the closely related, NNT-replete, B6N mouse to assess whether NNT or another strain-specific gene(s) causes differential phenotypes between substrains, as has been previously reported (13, 14, 15, 17).

[1]Centre for Endocrinology, William Harvey Research Institute, Charterhouse Square, Barts and The London School of Medicine and Dentistry, Queen Mary University of London, London, UK [2]Medical Research Council Harwell Institute, Mary Lyon Centre, Harwell Campus, Oxfordshire, UK [3]School of Human Sciences, London Metropolitan University, London, UK [4]William Harvey Heart Centre, William Harvey Research Institute, Charterhouse Square, Barts and The London School of Medicine and Dentistry, Queen Mary University of London, London, UK [5]Medical Research Council Harwell Institute, Mammalian Genetics Unit, Harwell Campus, Oxfordshire, UK

Correspondence: l.a.metherell@qmul.ac.uk

Here, we show that the B6J do not develop CM, even when aged to 18 mo, suggesting NNT loss alone does not cause CM. However, surprisingly over time, the B6N develop CM that is absent in both the B6J and B6J-Nnt. We go on to show that the CM in B6N may be due to a null mutation in myosin light chain kinase 3 (*Mylk3*), which is only present in B6N substrains. MYLK3 phosphorylates myosin light chain 2 (MYL2), which is itself essential for the assembly of actin fibres in the heart and mutated in many human cases of CM (19, 20).

# Results

## B6N mice develop CM, whereas B6J do not

To ascertain whether the B6J substrain developed CM, male mice from the three substrains B6N, B6J, and B6J-Nnt were analysed at 3, 12, and 18 mo of age, assessing cardiac form and function by echocardiography 1 wk before sacrifice (Fig 1 [12 mo], Fig S1 [collated], Fig S2 [3 mo], Fig S3 [18 mo], and Table S1 [collated]). This revealed a clear strain-dependent difference in dynamic heart structure, with dilated cardiomyopathy (DCM) in the B6N, particularly striking at 12 mo (Fig 1A and Table S1B). B6N mice had larger left ventricular internal diameter at both diastole and systole (Fig 1B and C) and correspondingly larger left ventricular volume (Fig 1D and E) than the B6J and B6J-Nnt. However, B6N mice had thinner left ventricular anterior wall and posterior wall thicknesses, indicating a thinning or stretching of the muscle walls (Fig 1F–I). This resulted in a lowered ejection fraction and reduced fractional shortening (Fig 1K and L). There was also a reduction in cardiac output, stroke volume, and heart rate in the B6N (Fig 1J, M, and N). We observed no difference in heart weight at this age (Fig 1O). These parameters represent a remarkable difference in the cardiac phenotype of these two closely related substrains consistent with a modest DCM in the B6N rather than the B6J mice.

At 3 mo, some aspects of the divergent phenotype were present, as the B6N hearts have higher left ventricular internal diameter and left ventricular volume at both systole and diastole, as well as thinner left ventricular posterior wall at systole than both B6J and B6J-Nnt mice (Table S1A and Fig S2B–N). At 18 mo, the phenotype maintains, as the B6N hearts had thinner anterior and posterior walls as well as enlarged ventricles (Table S1C and Fig S3B–N). The dilated B6N hearts were hypofunctional, reflected in the reduced ejection fraction and fractional shortening (Fig S3K and L). At 18 mo, there was far more intragroup variability (Table S1C), which may explain the slightly higher significance values at this age compared with 12 mo (Fig S1A–M).

Dilation of the ventricles and thinning of the walls in the B6N may indicate a difference in the heart size, commonly measured by heart weight. Although all three groups exhibited a progressive increase in heart weight as they got older (Figs 1O, S1N, S2O, and S3O), there was no significant difference amongst groups at any time point. Body size, assessed by tibia length, was no different between B6N and B6J substrains, indicating no change in the heart weight:tibia length ratio in the B6Ns (data not shown). We saw no difference in survival across the groups, indicating the phenotype is nonlethal under normal conditions.

## Cardiomyocytes are "stretched" in B6N hearts

Changes in cardiac function can be driven by gross structural changes such as those we see here, and this can be reflected in, and even driven by, changes in microscopic structure. Haematoxylin and eosin staining did not reveal any difference in heart size, as measured by the base–apex length (Fig 2A and B). We also observed no increased fibrosis in B6N hearts at 18 mo, barring one individual in the B6N group (Fig 2C and D). Interestingly, this individual was an outlier within this group in many echocardiographic parameters, having lower left ventricular volume and thicker muscle walls (Fig S3B).

Volume overload, similar to that observed in the B6N group, can stretch the ventricles and the heart responds by recruiting sarcomeres in series with existing sarcomeres, elongating the cardiomyocytes. This response, known as eccentric hypertrophy, is common in endurance athletes, but can also present in instances of cardiac dysfunction, and typically increases contractile force according to the Frank–Starling mechanism (21). We used wheat germ agglutinin to delineate cell membranes and allow us to measure cardiomyocyte length and cross-sectional area (CSA) in longitudinal and transverse sections, respectively (22) (Fig 2E and F). We observed an increase in cardiomyocyte length in B6N hearts, coupled with a striking reduction in CSA, suggesting the cardiomyocytes are "stretched" in these mice (Fig 2G and H). We hypothesised this might be due to changes in sarcomere organisation, typical of eccentric hypertrophy. We used MYL2 as a marker of the A band and alpha-actinin (ACTN1) as a marker of the Z-line in serial sections of heart tissue to measure sarcomere length (the distance between repeating units) and sarcomere thickness (across the short axis of the cell) (Fig 2I and J). B6N sarcomeres are thinner; however, there was no difference in sarcomere length (Fig 2K and L). Furthermore, there was no difference in the total expression of a panel of actomyosin and sarcomere markers when measured by Western blot and densitometry (MYL2, ACTN1, and TNNT2) (Fig 2M).

These data indicate both the form and function of the hearts in B6N mice are compromised. This may result from mutation of a key cardiac gene or may be acquired as a consequence of other disease states, such as hypertension, diabetes, or infection. However, we found no differences in blood biochemistry consistent with a hypertensive phenotype (Fig S4). There were no differences in plasma ions (Fig S4A); however, there were significantly higher total cholesterol, low-density lipoprotein, and high-density lipoprotein levels at 18 mo in blood from the B6N group but not at other time points (Fig S4B). Furthermore, other studies have shown the B6N mice in fact have lower blood pressure than the B6J, all but eliminating hypertension as the cause of the phenotype (23).

## B6N transcriptome is enriched for cardiac remodelling genes

Finding no obvious cause for the cardiac remodelling in the biochemistry or haematology, we wanted to determine whether changes in gene expression could underpin the differences in macro- and microstructure. To this end, we performed RNAseq on heart tissues from five 18-mo-old male mice per group. We used a CuffDiff TopHat pipeline to identify differentially expressed genes between groups. We initially performed pathway analysis on all

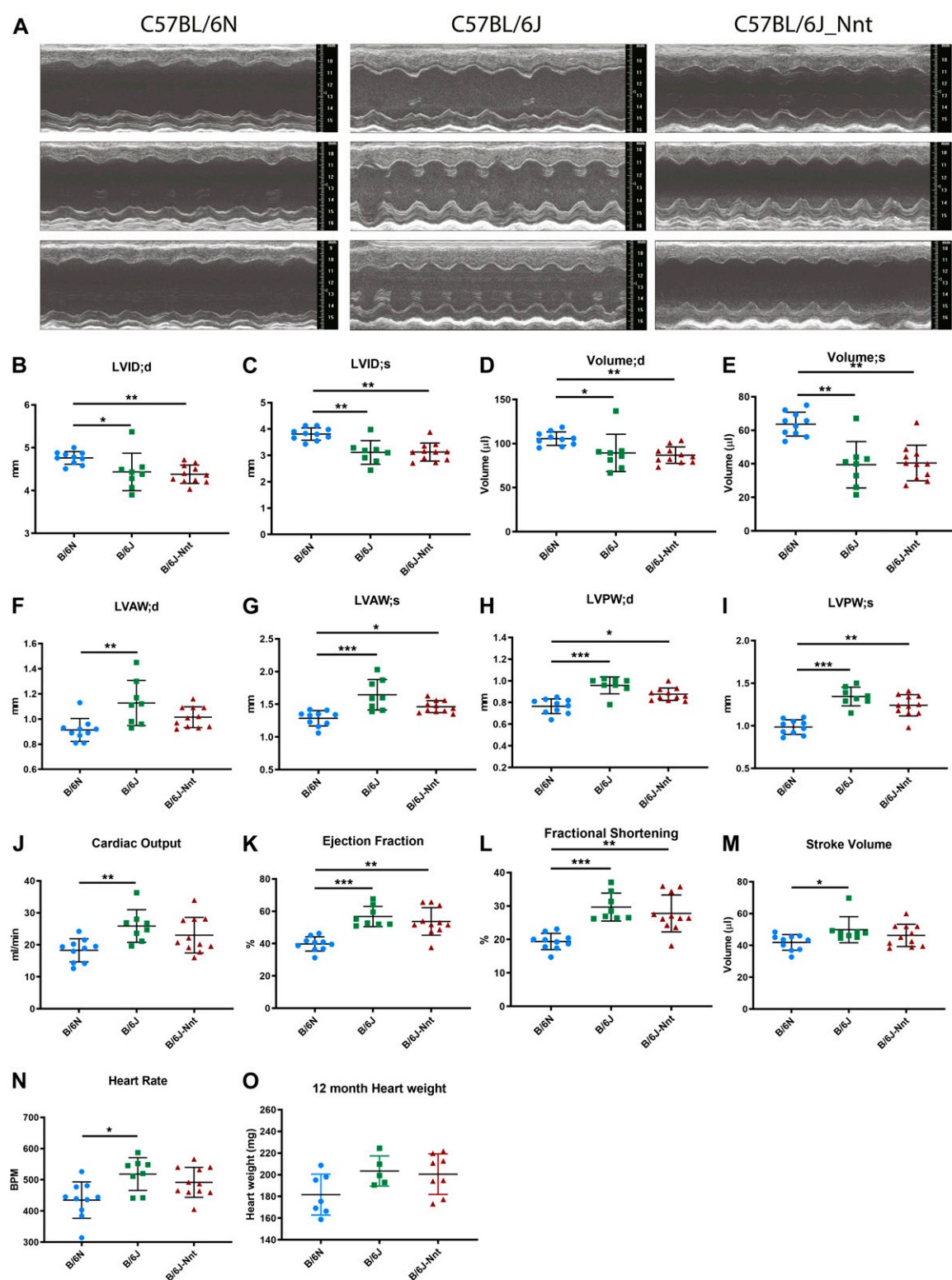

**Figure 1. B6N mice exhibit dilated CM.**
**(A)** Representative echocardiogram traces from three 12-mo-old mice at each genotype. **(B)** LVID;d, left ventricular internal diameter at the end of diastole. **(C)** LVID;s, left ventricular internal diameter at the end of systole. **(D)** Left ventricular volume at the end of diastole. **(E)** Left ventricular volume at the end of systole. **(F)** LVAW;d, left ventricular anterior wall thickness at the end of diastole. **(G)** LVAW;s, left ventricular anterior wall thickness at the end of systole. **(H)** LVPW;d, left ventricular posterior wall thickness at the end of diastole. **(I)** LVPW;s, left ventricular posterior wall thickness at the end of systole. **(J)** Cardiac output = stroke volume × heart rate. **(K)** Ejection fraction = stroke volume/ end diastolic volume. **(L)** Fractional shortening = (LVID;d − LVID;s)/LVID;d. **(M)** Stroke volume = volume at the end of diastole − volume at the end of systole. **(N)** Heart rate. **(O)** Heart weights of 12-mo-old mice at cull. Kruskal–Wallis, B6N n = 10, B6J n = 8, B6J-Nnt n = 11, mean ± SD, * < 0.05, ** < 0.01, *** < 0.001, **** < 0.0001.

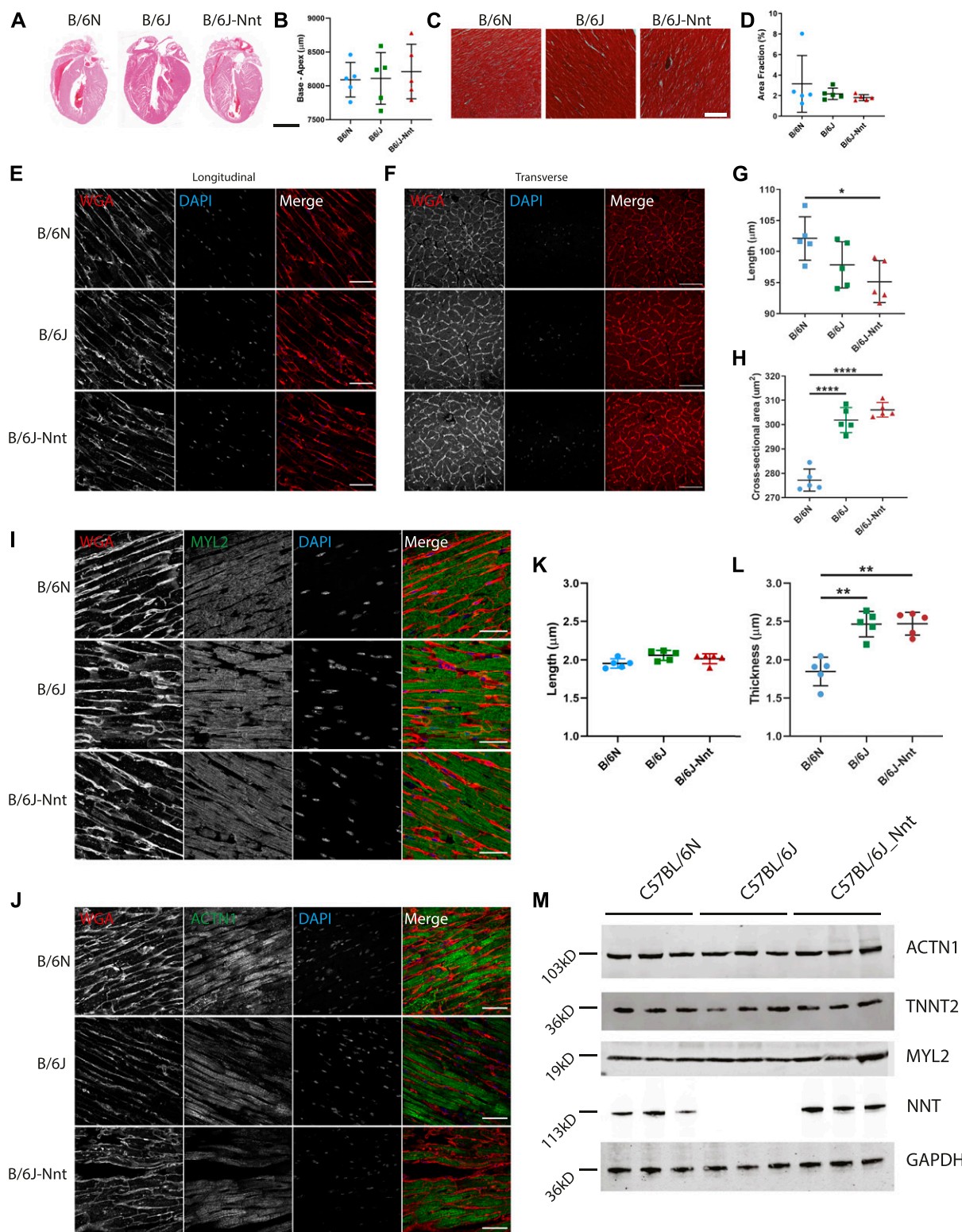

**Figure 2. Differences in microstructure of the heart amongst the three strains.**
**(A)** Representative images of 18-mo heart tissues stained with haematoxylin and eosin. Scale bar = 4,000 μm. **(B)** Base–apex measurements from H&E–stained sections. No significant difference in heart length. **(C)** Representative images of 18-mo heart tissues stained with Masson's Trichrome. Scale bar = 200 μm. **(D)** Quantitation of fibrosis from Masson's Trichrome–stained sections. n = 5 mouse hearts per group. Five sections per individual, three areas per section. Kruskal–Wallis test for grouped data, mean ± SD. **(E, F)** Immunofluorescent staining for wheat germ agglutinin (red) and DAPI (blue) in longitudinal (E) and transverse (F) sections of cardiomyocytes in heart tissues. Scale bar = 50 μm. **(E, F, G, H)** Quantitation of cardiomyocyte length (G) and cross-sectional area (H) from WGA staining in (E, F). Kruskal–Wallis test for

genes using Ingenuity Pathway Analysis. "Antigen presentation pathway" was the only pathway significantly up-regulated in the B6N compared with the B6J (Table S2), and there were no significantly enriched pathways between the B6N and B6J-Nnt, nor the B6J and B6J-Nnt (data not shown). We subsequently filtered the expression data for genes which were regulated in a strain-dependent manner (87 genes, Fig 3A and Table S3) and in an NNT-dependent manner (64 genes, Fig 3B and Table S4) (q < 0.05). Gene Ontology (GO) enrichment analysis revealed an overrepresentation of genes associated with muscle contraction in the strain-dependent analysis (Fig 3C) and numerous pathways associated with immune responses in NNT-dependent genes (Fig 3D).

Within the cluster of genes associated with muscle contraction, we found many genes typically associated with cardiac remodelling up-regulated in the B6N: myosin heavy chain 7 (*Myh7*), myosin light chain 4 (*Myl4*), myosin light chain 7 (*Myl7*), troponin T1, skeletal type (*Tnnt1*), natriuretic peptide precursor A (*Nppa*), and natriuretic peptide precursor B (*Nppb*). These genes are markers of hypertrophy and are often altered in the heart in disease states as the tissue reverts to a more "foetal" expression pattern 24, 25, 26. The expression levels of these genes and their associated binding partners were assessed in the three age-groups by RT-qPCR (Fig 4). Some of the differentially expressed cardiac genes were significantly altered on RT-qPCR (*Myh7 and Tnnt1*), whereas the others showed a strain-dependent trend. No sequence variants were found in these genes to explain their differential expressions (Table S5).

### Variant calling pipeline highlights Mylk3 as the likely causative gene

It is unclear whether the gene expression changes we see are drivers of or responses to the DCM phenotype. Almost all genes within the cardiac remodelling cluster have been linked to cardiomyopathies, not limited to the dilated form (27). We mined the RNAseq data for genetic variants between the B6N and B6J, seeking variants which might explain the differential gene expression patterns or be causal of the phenotype. Compared with the *Mus musculus* reference genome (B6J), 43 variants amongst the B6N and B6J substrains were identified, 38 in the B6N and 5 in the B6J, potentially affecting 63 genes (Table S5). A variant in *Mylk3*, Chr8:85365179A>T, c-5T>A, SNP ID rs245783224, was considered a good candidate because it is almost exclusively expressed in the heart, with ablation resulting in cardiac dysfunction, which is modest in the germ line knockout (28, 29, 30). In humans, it is highly expressed in heart and ~10-fold lower in skeletal muscle, with two recent articles showing an association of variants in *MYLK3* with DCM (31, 32).

### Point mutation in Mylk3 abolishes protein expression in B6N mice

We confirmed this mutation was present in B6N and absent in B6J mice by Sanger sequencing (Fig 5A) and that this had no effect on mRNA expression (Fig 5B). Of the 36 inbred mouse strains sequenced by the Sanger Mouse Genomes Project (33), only the C57BL/6NJ mouse has the variant. The mutation introduces a putative translation initiation site (TIS), 5-bp upstream of the canonical TIS. We used a series of Kozak signature prediction software to assess whether the new TIS created by the mutation would be a better match for translation initiation than the canonical TIS (Fig 5C) (34, 35, 36). TIS miner and NetStart 1.0 predicted the variant weakens the canonical TIS and PreTIS predicted the new ATG to form an alternative TIS with a very high confidence score (0.93) (37). The predicted protein product from this novel TIS would be out of frame and result in an altered protein with a molecular weight of 3.2 kD. In keeping with this, MYLK3 protein expression in the B6N mice was absent by Western blot (Fig 5D). It is important to note MYLK3 expression was absent at all three ages, showing the "genotype" of these mice with respect to MYLK3 was consistent across all groups (Fig S5C and D). We also performed immunohistochemistry (IHC) on formalin-fixed paraffin-embedded samples (Fig S5A and B), finding no MYLK3 expression in the C57BL/6N mice. Intriguingly, we found a gradient of staining in the C57BL/6J mice, with high staining in the papillary muscles and at the midline of the tissue and less in the endocardium and at the base and apex. This was surprising, as it is in stark contrast to the previously reported phospho-MYL2 gradient (19).

To verify the deleterious effect of this variant on protein translation, a luciferase reporter assay was performed using vectors containing either the wild-type (B6J) or variant (B6N) 5′ UTRs of *Mylk3* transcript variant 1 (NM_175441.5), immediately upstream of the luciferase start codon. Luciferase signal was normalised by co-transfecting with a GFP vector and normalising luminescence to fluorescence (Fig 6A). Luciferase activity was significantly lower in HEK293T cells transfected with the "B6N-5′UTR-*Mylk3*-Luc" vector than those transfected with "B6J-5′UTR-*Mylk3*-Luc," indicating the mutation is sufficient to abrogate protein translation (Fig 6B). This was confirmed by expressing MYLK3 constructs with the respective 5′ UTRs under a T7 promoter in a cell-free protein expression system (Promega). The B6J-5′UTR-*Mylk3*-Luc construct produced a protein at ~86 kD, whereas B6N-5′UTR-*Mylk3*-Luc showed negligible expression (Fig 6C) and no other protein was detected. This ~86-kD protein was confirmed to be MYLK3 by Western blotting (Fig 6D).

We analysed the other three myosin light chain kinase genes, suspecting one of these may compensate for the absence of MYLK3 in the B6N mouse. RNAseq data revealed *Mylk* and *Mylk2* expression levels were very low in the heart; however, *Mylk4* expression level was similar to *Mylk3*. Interestingly, *Mylk4* expression was higher in the B6N mice and was identified as a strain-dependent gene by RNAseq (Table S2). MYLK4 could phosphorylate key targets in the absence of MYLK3 in the B6N, and this may allow the mice to survive to adulthood. RT-qPCR revealed *Mylk4* was ~1.8-fold higher in the B6N group at both 3 and 12 mo; however, there was no difference in protein expression and is, therefore, unlikely to be able to compensate for loss of MYLK3 (data not shown).

---

grouped data, n = 5, mean ± SD. **(I)** Immunofluorescence staining for WGA (red), MYL2 (green), and DAPI (blue). Scale bar = 50 μm. **(J)** Immunofluorescence staining for WGA (red), alpha-actinin (ACTN1) (green), and DAPI (blue). **(I, J, K, L)** Quantification of sarcomere length and thickness from images in (I) and (J) using MyofibrilJ. n = 5 mouse hearts per group. Five sections per individual, three areas per section. Kruskal–Wallis test for grouped data, mean ± SD. **(M)** Western blot for ACTN1, TNNT2, MYL2, GAPDH, and NNT in 18-mo heart lysates, n = 3 mice per group. * < 0.05, ** < 0.01, *** < 0.001, **** < 0.0001.

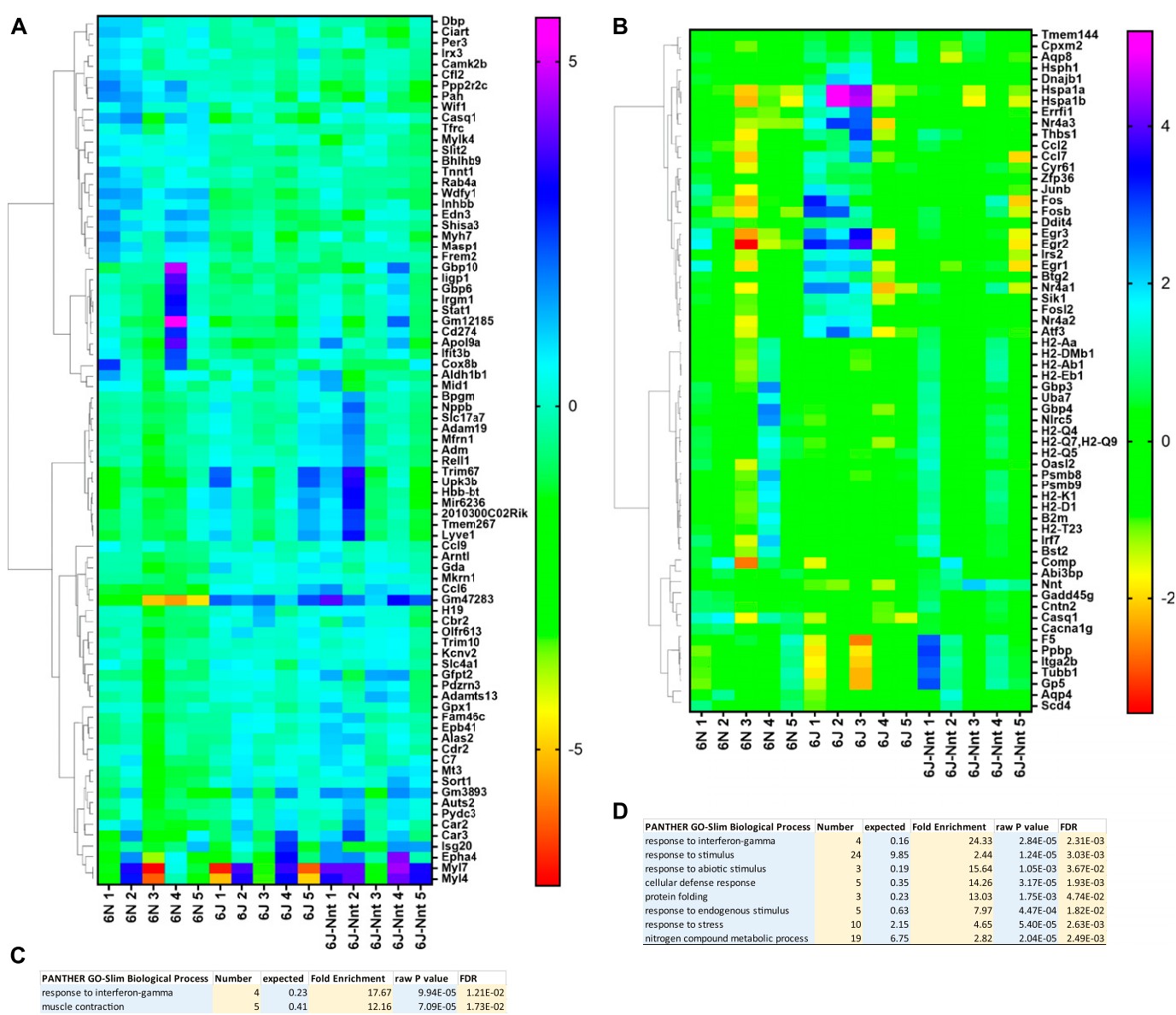

**Figure 3. RNAseq reveals strain-dependent and NNT-dependent genes.**
Differentially expressed genes generated by CuffDiff TopHat pipeline, clustered according to the expression profile across samples. Values for each cell are calculated as the log₂[x + 1], where x is the counts for that gene. Each subsequent value for a gene is then subtracted from the average of all the values for that gene. These values are then used to generate the heat map. Mouse group indicated at the base of each column. **(A)** Heat map of gene list segregated by strain. **(B)** Heat map of gene list segregated by NNT status. **(A, C)** Gene Ontology list generated by Panther open access software from gene list in (A). **(B, D)** Gene Ontology list generated by Panther open access software from gene list in (B).

# Discussion

Since the discovery of the unique *Nnt* mutations in B6J mice, many groups have used this mouse in conjunction with the B6N to study the role of *Nnt*, as one might with a traditional knockout and corresponding wild-type model (13, 38, 39, 40, 41, 42 43). In contrast to previous studies linking *Nnt* to cardiac function (12,13,14,16), we show that *Nnt* expression levels do not correlate with the DCM phenotype, although a number of gene expression profiles tracked with *Nnt* status in our mouse models (Fig 3B). Instead, we identify a mutation in *Mylk3* in B6N mice that abolishes protein expression

and likely causes DCM, whereas MYLK3-replete B6J mice are unaffected. Interestingly, the same *Mylk3* variant (g.chr8:85365179A>T; c.-5T>A) was previously described and, although that study found no evidence of DCM at an early time point (16 wk), it did show the B6N heart papillary muscles contracted with less force than those from B6J hearts (44).

Here, we show B6N cardiomyocytes are "stretched," reflected by a thinning of the sarcomeres within each cell but found no difference in individual sarcomere length. The latter finding is particularly interesting, as one might expect volume overload to stretch the muscle fibres and increase the sarcomere length. Increases in sarcomere

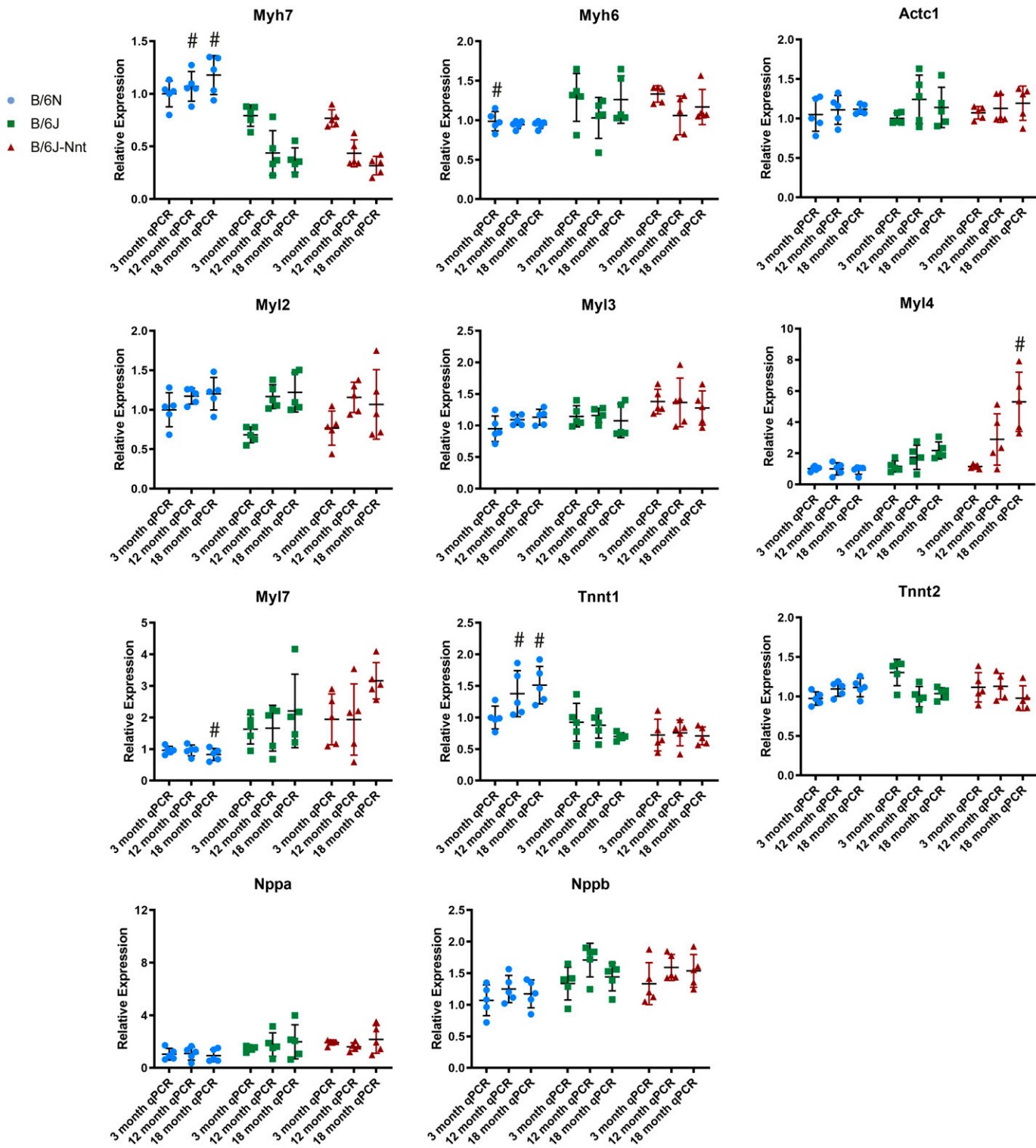

**Figure 4. RT-qPCR analysis of key cardiac genes.**
Collated RT-qPCR data. All RT-qPCR expression values generated by ΔΔCT method. Actc1, actin, alpha, cardiac muscle 1; Myh6, myosin heavy chain 6; Myh7, myosin heavy chain 7; Myl2, myosin light chain 2; Myl3, myosin light chain 3; Myl4, myosin light chain 4; Myl7, myosin light chain 7; Nppa, natriuretic peptide precursor A; Nppb, natriuretic peptide precursor B; Tnnt1, troponin T1, skeletal type; Tnnt2, troponin T2, cardiac type. RT-qPCR expression of key cardiac genes. Kruskal–Wallis test at each time point, n = 4, mean ± SD. # symbol indicates one substrain differs significantly from the other two at that time point with a *P*-value less than 0.05.

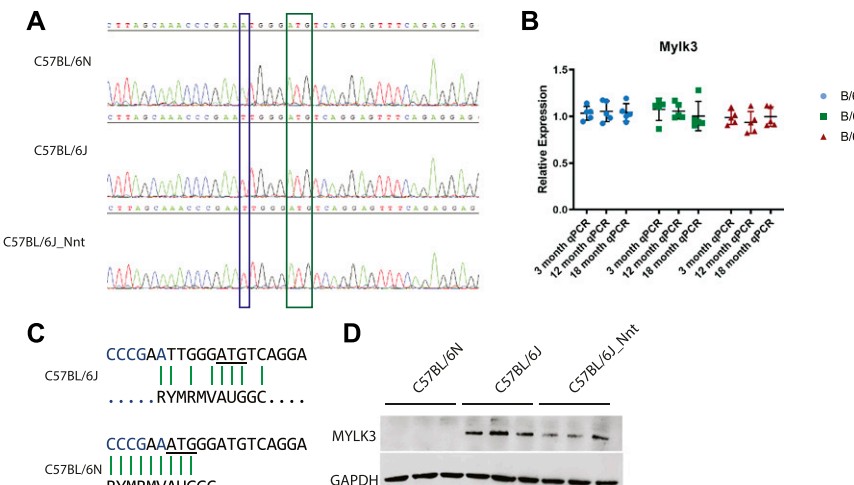

Figure 5. SNP in *Mylk3* abolishes protein expression in C57BL/6N hearts.
**(A)** Representative chromatogram of Sanger sequencing of *Mylk3* in DNA isolated from 3-mo heart samples of each group. Blue box indicates SNP and A of alternate TIS in B6N and green box indicates the canonical translation initiation site. **(B)** RT-qPCR analysis of *Mylk3* expression. One-way ANOVA at each time point; n = 5, mean ± SD. **(C)** Kozak signature analysis of translation initiation site in B6J (upper pane) and B6N (lower pane) showing the alternate TIS for the B6N variant. **(D)** Western blot staining of GAPDH and MYLK3 in 3-mo heart tissue lysates.

length enhance the calcium sensitivity of the fibres and increase contractility (45). The lack of significant difference in sarcomere length may be due to the comparatively small difference in cardiomyocyte length. It is unclear what the effect of sarcomere narrowing may be on contractility. It may increase the frequency of actin and myosin crossbridge formation; however, the evidence for this is unclear (46, 47).

*Mylk3* is a very plausible candidate for the genesis of the DCM seen in B6N mice. A previous study generated *Mylk3*⁻/⁻ mice and showed they develop a similar phenotype of dilated ventricles, reduced fractional shortening, and reduced ejection fraction, whereas

their phenotype is similarly nonlethal (28). As the cardiac isoform of the family of myosin light chain kinases, MYLK3 phosphorylates MYL2 and potentially autophosphorylates itself (44, 48). MYLK3 expression is also important for sarcomere assembly. Knockdown of MYLK3 impaired sarcomere assembly in cultured cells and in zebra fish (49). Conditional adult-onset *Mylk3* knockout mice have drastically reduced MYL2 phosphorylation which precedes heart failure (29, 30), whereas germ line ablation causes mice to develop DCM in adulthood, and perinatal ablation produces an intermediate phenotype (28). A previous report demonstrated B6N mice have reduced rather than absent MYL2 phosphorylation (44) because of absence of MYLK3,

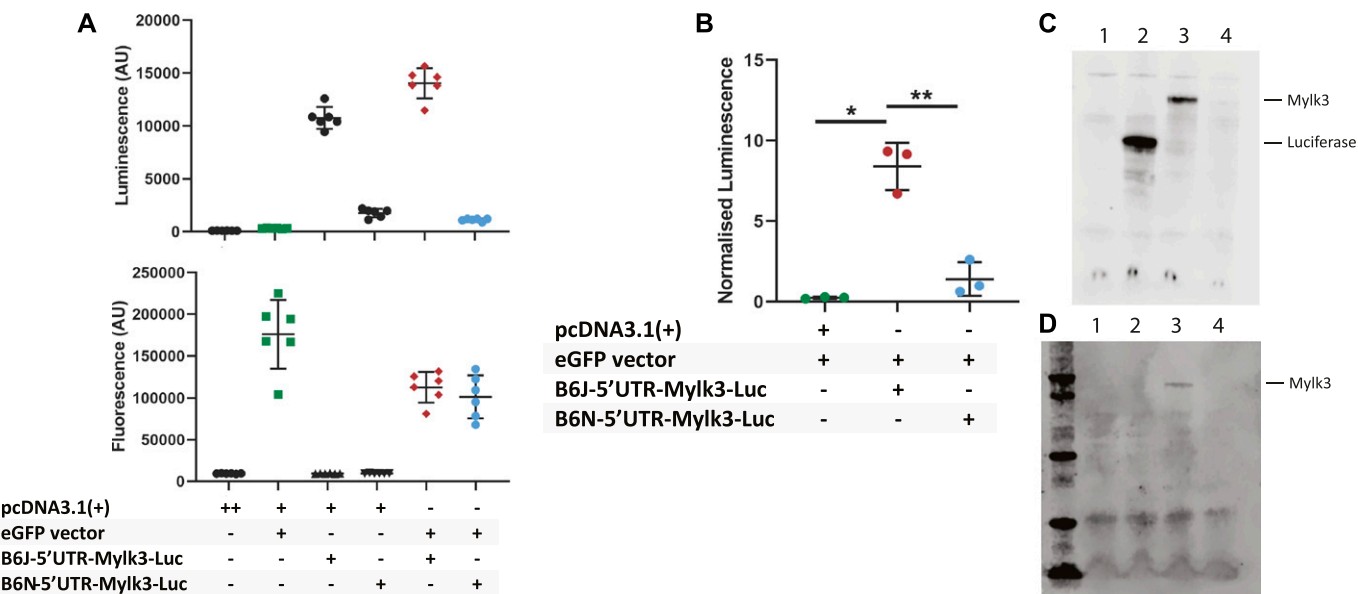

**Figure 6. 5′UTR MYLK3 mutation in C57BL/6N is sufficient to abrogate protein expression.**
**(A)** Luminescence (upper panel) and fluorescence (lower panel) signal in cells transfected with a combination of empty vector, GFP, and luciferase vectors signified by the table. Representative experiment from three experimental repeats with six technical replicates. **(B)** Normalised luminescence. One-way ANOVA Brown–Forsythe and Welch test; n = 3, mean ± SD. **(C)** Coupled transcription–translation expression of genes under control of T7 promoter: Sample in lane 1 contains no template DNA; lane 2 contains a luciferase control vector; lane 3 contains a pcDNA3.1(+) vector expressing MYLK3 with the 5′UTR from C57BL/6J mice; lane 4 contains a pcDNA3.1(+) vector expressing MYLK3 with the 5′UTR from C57BL/6N mice. Proteins detected by chemiluminescent substrate. **(D)** Samples as above, detected by blotting with MYLK3. **(C, D)** are representative of three repeats.

but it has been postulated that another myosin light chain kinase family member could be maintaining the residual phosphorylation seen (50).

Phosphorylation of MYL2 is important for the assembly of actin fibres, and overexpression of MYLK3 increases the speed of the contractions and relaxations of a spontaneously beating cardiac cell culture model (19, 48). MYL2-knockout mice are embryonic lethal (51), yet MYL2 phosphomutants (S14A and S15A) survive to adulthood, develop DCM and, although more than half survive beyond 12 mo, die at a significantly quicker rate (52). The disparity in survival between these different mutants and the germ line versus adult-onset MYLK3 knockouts is worth remarking. Given the lethality of conditional adult MYLK3 knockouts, it has been suggested that the lack of MYLK3 in germ line knockouts during development allows the heart to adapt, whereas this ability is lost when the knockout is induced in adulthood, as these mice develop heart failure and die (28, 29, 30). Intriguingly, we found up-regulation of *Mylk4* expression, present at relatively high levels in the heart, but this up-regulation was absent at the protein level. It is likely another unidentified protein is compensating for the loss of MYLK3 during development, facilitating survival and limiting disease progression. Discovering the identity of this kinase will be important to future work into the role and of MYLK3, in both the broader context of its activity and for the C57BL/6N mouse.

The lack of MYLK3 may also explain the differential responses to stresses such as transverse aortic constriction (TAC) and angiotensin II treatment observed in other studies comparing C57BL/6 mouse substrains (15, 17). The B6N substrains had dilated ventricles, thinner muscle walls, and increased expression of foetal gene *Nppb* compared with a B6J substrain after TAC (17) and showed increased fibrosis and foetal gene expression in response to chronic angiotensin II treatment (15). Interestingly, B6N mice die at a far greater rate than B6J mice after TAC (17). This mimics the increased mortality in *Mylk3$^{-/-}$* mice compared with WT littermates after TAC (28). We believe the increased mortality in B6N mice is due to the *Mylk3* mutation present in this substrain.

In our RNAseq data, we find B6N hearts express four times more *Myh7* than hearts from the two B6J substrains. *Myh7* is one of two main cardiac myosin heavy chain isoforms, the other being myosin heavy chain 6 (*Myh6*) (53). In humans, MYH7 is predominant in the ventricles, whereas MYH6 is predominant in atria (54, 55). In mice, this pattern is observed only during embryogenesis, and then MYH6 is the main isoform across the whole heart (56). Although believed to have arisen from a duplication event, the two isoforms have distinct sequences and activities as myosin heavy chain proteins: MYH6 is a "fast-twitch" cardiac isoform, with faster binding kinetics and contraction speed than MYH7 (24). Isolated papillary muscles from B6N mice contract more slowly and with less force and relax slower than the equivalent muscle isolates from B6J mice.

However, it is equally possible that the increase in *Myh7* in B6N reflects a stress adaptation of the tissue due to the dilation, particularly as the RNAseq was performed on 18-mo heart tissues. Under conditions of stress, transcription of foetal genes such as *Myh7* are often increased and this is thought to be due to a de-repression of their expression by microRNAs (25, 26, 57) The picture is not quite so clear in our case, as the levels of other foetal genes, particularly *Nppa* and *Nppb*, are higher in the B6J substrains. This may reflect a more complicated regulation by distinct microRNAs,

but we believe the difference in *Myh7* expression is a response to the dilation of the ventricles, rather than its genesis.

The C57BL/6 strain is the most widely used laboratory strain, initially made popular for its strong breeding characteristics; the B6J substrain, originating in The Jackson Laboratory, is by far the most widely used substrain, accounting for 90% of all published references to the C57BL/6 strain. However, over the past decade, international mouse knockout initiatives, the IMPC, European Conditional Mouse Mutagenesis (EUCOMM), and the National Institutes of Health Knockout Mouse (KOMP) have moved to the C57BL/6N as a background strain because of greater ES cell efficiency for homologous recombination (58, 59).

It is becoming increasingly clear that these two closely related substrains have many more genetic/phenotypic differences than is widely appreciated. Worryingly, one review found that in the journal *Diabetes*, 63% of all published articles between 2010 and 2014 that used a variant of the C57BL/6 strain and made no mention of the exact substrain used (6), instead simply stating C57BL/6 mice were used. This appears to be true with other mouse strains, as the authors found 58.5% of all methods sections did not report the exact substrain with sufficient completeness. Aided by next-generation sequencing approaches, there are now sizeable databases documenting the many genetic differences between mouse substrains, including those involving the B6J and B6N (23, 33, 60, 61, 62). These produce remarkably different phenotypes regarding weight, blood pressure, bone density, blood glucose, and cardiovascular traits (3, 17, 23, 63, 64, 65, 66, 67, 68). In this regard, using the partly compromised B6N model, phenotypic effects may be more apparent and may help us unveil new important cardiac genes, and hence new therapies. However, caution must be advised, as researchers may want to avoid labelling effector genes as "causative" of CM or "essential" for cardiac function when made on the 6N background.

## Conclusions

We have described a *Mylk3* mutation that is probably responsible for a modest CM in the C57BL/6N substrain and demonstrated that NNT ablation in the C57BL/6J mouse does not, by itself, cause a cardiac phenotype. The *Mylk3* mutation is absent in all other mouse substrains sequenced in the Mouse Genomes Project including the closely related C57BL/6J mouse. However, many other genomic variants exist between substrains, the consequences of which are only gradually being uncovered (61, 69, 70). We clearly demonstrate how a single gene defect can cause susceptibility to a particular phenotype in a mouse substrain, in this case CM in the B6N. Hence, this study accentuates the importance, in a wider setting, of knowing the genetic defects in your mouse substrain and highlights the value of selecting your mouse background with care.

# Materials and Methods

### Animal husbandry

All mice were bred, housed, and culled at Medical Research Council Harwell, and therefore, the husbandry was identical for all three

substrains. Only male mice were used in this study. Mice were kept and studied in accordance with the UK Home Office legislation and local ethical guidelines issued by the Medical Research Council (Responsibility in the Use of Animals for Medical Research, July 1993; Home Office license 30/3146). Mice were kept under controlled light (light 7 AM–7 PM, dark 7 PM–7 AM), temperature (21°C ± 2°C), and humidity (55% ± 10%) conditions. They had free access to water (9–13 ppm chlorine) and were fed ad libitum on a commercial diet (SDS Rat and Mouse No. 3 Breeding diet, RM3, 3.6 kcal/g). The mouse strains used were C57BL/6NTac originally from Taconic (Taconic Biosciences), C57BL/6J distributed by Charles River (Charles River UK), and C57BL/6J mice carrying a bacterial artificial chromosome transgene to restore murine *Nnt* (officially C57BL/6J-Tg(RP22-455H18)TG1Rdc/H). All mice used were no more than 10 generations of mating removed from restock.

### Echocardiography

A week before culling, the cardiac phenotype of each cohort of mice was assessed using echocardiography. The mice were anaesthetised under 4% isoflurane, then maintained at 1.5–2% for the duration. For the echocardiogram, the left ventricle of the heart was imaged and analysed using the Vevo770 (FUJIFILM VisualSonics) according to the IMPC phenotyping protocol. More details of this procedure can be found on www.mousephenotype.org/impress.

### Clinical chemistry

Free-fed mice were terminally anaesthetised at 3, 12, or 18 mo with isoflurane, and blood was collected from the retro-orbital sinus using uncoated glass capillary tubes (Cat. no. 612-2439; VWR) into either lithium heparin or EDTA-coated paediatric tubes (Cat. no. 078028 & 077030; Kabe Labortechnik). Lithium heparin samples were kept on wet ice and centrifuged within 1 h of collection for 10 min at 5,000$g$ in a refrigerated centrifuge set to 8°C. Plasma samples were analysed for sodium, potassium, chloride, glucose, triglycerides, free fatty acids, and total-, high-density lipoprotein–, and low-density lipoprotein cholesterol on board a Beckman Coulter AU680 clinical chemistry analyser using reagents and settings recommended by the manufacturer and as per the IMPC phenotyping pipeline (http://www.mousephenotype.org/impress/procedures/7).

### Tissue isolation

After bleeding, mice were dissected under a laminar flow tissue culture hood. Tissues were removed and immediately snap frozen in liquid nitrogen or immersed in 4% PFA. All animals were culled between 1,000 and 1,130 h. All animal protocols in this study were approved by the United Kingdom Home Office.

### Heart weights and tibia lengths

Frozen isolated heart tissues were weighed to an accuracy of 0.1 mg. Tibia lengths of C57BL/6N and C57BL/6J mice were measured using X-ray at 13 wk.

### Genotyping

Genomic DNA was extracted from the mouse tail tissue using a DNeasy tissue kit (QIAGEN). Mice were genotyped for *Nnt* status using previously published primers (3).

### Tissue extraction for RNA-Seq

Five 18-mo-old mice of the three different substrains were used per group. RNA was isolated from intact hearts using the RNeasy Mini Kit (QIAGEN). RNA samples were processed by INVIEW Transcriptome Discover service (GATC Biotech) and resultant FASTQ files processed as follows.

### Quality filtering

We used Trim Galore (http://www.bioinformatics.babraham.ac.uk/projects/trim_galore/), a wrapper tool around Cutadapt (71) and FastQC (72) (https://www.bioinformatics.babraham.ac.uk/projects/fastqc/), to filter reads with quality Phred score cutoff less than 20 and trim Illumina TruSeq adapters in FASTQ files.

### Differential gene expression

We used TopHat (73) to align quality filtered reads to the *M. musculus* reference genome (GRCm38) (74), discarding final read alignments having more than two mismatches. We obtained 89.0%, 88.8%, and 87.8% concordant pair alignment rate for heart knockout, wild-type, and rescue, respectively. Finally, we used CuffDiff (75) to calculate expression in samples and test the statistical significance of each observed change in expression between them, using a q-value of <0.05 as the threshold for significance.

### RNAseq variant calling

We followed GATK pipeline for variant calling in RNAseq (https://software.broadinstitute.org/gatk/documentation/article.php?id=3891). We used the STAR 2-pass method from STAR aligner (76, 77) to align the quality filtered reads to the reference genome downloaded from ftp://ftp-mouse.sanger.ac.uk/ref/GRCm38_68.fa. Then, we used Picard to add read groups, sort, and mark duplicates (https://broadinstitute.github.io/picard/). We used SplitNCigarReads from GATK (78, 79, 80), to split reads into exon segments and hard-clip any sequences overhanging into the intronic regions. We then applied indel realignment and base recalibration from GATK. Finally, we used HaplotypeCaller (81 *Preprint*) for variant calling. Variants were validated by visualising alignments in the Integrative Genomics Viewer browser (82) from the Broad Institute.

Compared with the *M. musculus* reference genome (6J), 79 variants amongst the B6N and B6J substrains were identified, potentially affecting 98 genes. Visual inspection of these variants in Integrative Genomics Viewer browser allowed us to eliminate 35 variants on the basis (mostly) of misalignment of short-read sequencing in repetitive regions, leaving 43 variants which differed from the B6J reference sequence with the potential to affect 63 genes. 38 of these varied in the B6N substrain with the

remaining five in B6J, all were biallelic with the exception of one variant, in cue domain-containing protein 1 (Cuedc1) which was monoallelic in B6J. This apparent heterozygosity is likely due to alignment of short sequencing reads to more than one locus.

Amongst the biallelic variants, the one in *Mylk3*, Chr8:85365179A>T, c-5T>A, SNP ID rs245783224, was considered a likely candidate because MYLK3 is almost exclusively expressed in the heart, with ablation resulting in cardiac dysfunction, which is modest in the germ line knockout (28, 29, 30). In humans, it is highly expressed in heart with two recent articles showing an association of variants in MYLK3 with DCM (31, 32).

### GO-term analysis

Gene lists were analysed using Panther Gene List analysis and a statistical overrepresentation test. GO terms for pathways overrepresented in these lists were generated by the software (83, 84).

### RT-qPCR

We isolated RNA from whole tissues using the TRIzol method, followed by a sodium acetate–ethanol cleanup to remove salt contamination (85, 86). Genomic DNA was removed by comprehensive DNAse treatment and RNA was converted to cDNA (Invitrogen). cDNA was then used to assess RNA expression by qPCR, calculated by the ΔΔCT method (Kapa Biosystems). Amplicons obtained from manually designed primers were confirmed by Sanger sequencing (GATC).

Primers were manually designed using PrimerQuest Tool (Integrated DNA Technologies) and synthesised by Sigma-Aldrich (Merck). Primer sequences are listed in the following table.

**Individual gene names with corresponding forward and reverse primer sequences for RT-qPCR. Sequences are all expressed 5′ –> 3′**

| Gene | Forward primer sequence | Reverse primer sequence |
|------|------------------------|-------------------------|
| *Gapdh* | GCCTTCCGTGTTCCTACC | CCTGCTTCACCACCTTCTT |
| *Myh7* | CTCAAGCTGCTCAGCAATCTA | GACACGGTCTGAAAGGATGAG |
| *Myh6* | GGATATTGATGACCTGGAGCTG | AGCCATCTCCTCTGTTAGGT |
| *Actc1* | AGCCCTCTTTCATTGGTATGG | CCTCCAGATAGGACATTGTTGG |
| *Myl2* | AGAAAGCCAAGAAGCGATAG | CTCTGTTCTGGTCCATGATTGT |
| *Myl3* | GGGCGAGATGAAGATCACATAC | TGGAATTGAGCTCTTCCTGTTT |
| *Myl4* | AACCCAAGCCTGAAGAGATG | TCCACGAAGTCCTCATAGGT |
| *Myl7* | CGTGGCTCTTCTAATGTCTTCT | CAGATGATCCCATCCCTGTTC |
| *Tnnt1* | GCCCTTGAACATCGACTACA | TCAACTTCTCCATCAGGTCAAA |
| *Tnnt2* | GGAGAGAGAGTGGACTTTGATG | CTTCCTCCTTCTTCCTGTTCTC |
| *Nppa* | TTTGGCTTCCAGGCCATATT | CATCTTCTACCGGCATCTTCTC |
| *Nppb* | ACTCCTATCCTCTGGGAAGTC | GCTGTCTCTGGGCCATTT |
| *Mylk3* | GAATTCCAAGGTGGCTGATTTC | TCAGTACATGGTTTGGCTTCA |

### Western blotting

Protein lysates were obtained by homogenising and lysing tissues in radio immunoprecipitation assay buffer for 2 h at 4°C. The lysates were sonicated at 20 kHz twice for 10 s, then centrifuged at 16,000$g$ for 15 min, and the resultant supernatants were normalised for total protein concentration with a bicinchoninic acid assay. Protein lysates were added to an equal volume of Laemmli buffer. Lysates were heated at 95°C for 5 min and then loaded into 4–12% SDS gels. Immunoblots for NNT were performed with unheated samples. Proteins were then separated based on size using electrophoresis and transferred to a nitrocellulose membrane (Sigma-Aldrich) by semidry transfer blot (Bio-Rad) at 15 V for 1 h. Membranes were probed with a target antibody and GAPDH. Antibodies used are as follows: mouse anti-NNT (1:1,000, HPA004829; Sigma-Aldrich), mouse anti-TNNT2 (1:2,000, Cat. no. ab8295, RRID:AB_306445; Abcam), mouse anti-Sarcomeric Alpha Actinin (1:2,000, Cat. no. ab9465, RRID: AB_307264; Abcam), rabbit anti-MYLK3 (1:1,000, Cat. no. AP7965a, RRID:AB_2266783; Abgent), rabbit anti-GADPH (1:2,000, Cat. no. ab9485, RRID:AB_307275; Abcam), and rabbit anti-MYL2 (1:5,000, Cat. no. ab79935, RRID:AB_1952220; Abcam).

### H&E

Freshly isolated hearts were immersed in 4% PFA on ice overnight. Tissues were then dehydrated and embedded in paraffin wax. We cut 7-$\mu$m sagittal sections through the tissues, making note of serial section number. Sections obtained at similar planes were then used for haematoxylin and eosin staining using a Leica Autostainer (87). Briefly, tissue sections were washed in distilled water followed by staining with Gill's 3 Hematoxylin to stain nuclei and Eosin Y to counterstain cytoplasmic regions. We imaged stained sections using a Panoramic 250 (3D Histech) slide scanner with CaseViewer software.

### Masson's Trichrome

We stained paraffin-embedded heart sections as described in reference 88. Briefly, we fixed sections in Bouin's solution for 1 h followed by intense washing. We then stained fixed sections with Weigert's haematoxylin (nuclei), Biebrich scarlet-acid fuchsin solution (plasma), and Aniline Blue (collagen). We imaged stained sections using a Panoramic 250 (3D Histech) slide scanner with Panoramic Viewer software. We assessed fibrosis in these images in ImageJ using a colour threshold–detecting blue staining, determined by a prenominated window for colour hue, and passing a fixed intensity threshold.

### Immunohistochemistry

We used two detection methods for protein localisation, one using chromogenic diaminobenzidine oxidation assays and the other epifluorescence coupled with conjugate fluorescent dyes. Chromogenic diaminobenzidine staining of MYLK3 was performed by a Ventana automated staining station using the OmniMap Rabbit HRP (Roche). We imaged stained sections using a Panoramic 250 (3D Histech) slide scanner with CaseViewer software.

For epifluorescence staining, paraffin tissue sections were cleared and dehydrated followed by boiling in a citrate buffer to expose protein epitopes. Specimens were then blocked in a BSA–serum mixture, followed by primary antibody incubation overnight. Samples were washed and incubated with secondary antibody, washed and finally mounted in an aqueous mountant. For immunofluorescent sections, tissues were counterstained with DAPI and wheat germ agglutinin Alexa Fluor 555 conjugate (Thermo Fisher Scientific) before background removal with TrueView Autofluorescence quenching kit (Vector Labs) followed by mounting in VectaShield mounting medium. Fluorescent slides were imaged on a Zeiss LSM 880 with Airyscan and quantified by MyofibrilJ package in ImageJ. DAB-stained slides were imaged using a Panoramic 250 (3D Histech) slide scanner with CaseViewer software. For sarcomeric morphometrics, longitudinal and transverse heart sections were used to measure the long axis (longitudinal sections) and CSA (transverse section) of individual cardiomyocytes delineated by WGA, alpha-actinin (1:5,000, Cat. no. ab9465, RRID:AB_307264; Abcam), and MYL2 (1:1,000, Cat. no. ab79935, RRID:AB_1952220; Abcam) staining. Confocal microscopy was performed with an LSM880 Microscope (Zeiss) on a 63× objective.

### Luciferase assay

The 5' UTR of *Mylk3* transcript variant 1 from B6N or B6J mice was cloned immediately upstream of the luciferase ORF in PGL3 (Promega) by Gibson cloning (89) from a gBlock (Integrated DNA Technologies) to create the reporter constructs "B6N-5'UTR-Mylk3-Luc" and "B6J-5'UTR-Mylk3-Luc." A CMV promoter and enhancer was also substituted for the SV40 promoter to increase expression. All constructs were verified by sequencing. 50 ng of reporter vector or control vector (pcDNA3.1(+)) was co-transfected with 50 ng pcDNA3.1(+) expressing GFP using lipofectamine 3000 reagent (Invitrogen). 24 h post-transfection the cells were incubated with a luciferase recording medium containing D-luciferin at 0.45 mg/ml and bioluminescence was recorded in an integrated ClarioStar microplate reader (BMG Labtech) for 30 min. The plate was washed three times in PBS, and GFP fluorescence was recorded. Luciferase activity was normalised to fluorescence to control for transfection efficiency. The experiment was performed three times with six technical repeats per experiment.

### Coupled transcription/translation assay

A MYLK3 ORF clone generated by GenScript (Clone ID: OMu23765) in the pcDNA3.1(+) vector was used as the backbone for subsequent 5'UTR cloning. 5' UTR of B6N and B6J MYLK3 was cloned immediately upstream of the canonical ATG and immediately downstream of the T7 promoter by Gibson Assembly (89) and confirmed by Sanger sequencing. 1 $\mu$g of plasmid was added to 40 $\mu$l of TnT Quick Master Mix (Promega) with 1 $\mu$l of methionine and 1 $\mu$l of Transcend Biotin-Lysyl-tRNA and incubated at 30°C for 1 h. A negative control reaction was set up containing no plasmid DNA, and a positive control containing a luciferase expression vector. The products of each reaction were separated by SDS–PAGE and then transferred onto a nitrocellulose membrane. The membrane was blocked in 5% BSA and incubated with MYLK3 and detected by Western blot protocol or with streptavidin–HRP conjugate and incubated for 2 h, followed by

detection with substrate as per the manufacturer's instructions (Promega). Membranes were visualised using a Licor Odyssey Fc and ImageStudio software.

### Statistical analysis

Statistical tests are described on a per experiment basis in the figure legends describing the experiment in question.

### Availability of data and materials

The dataset supporting the conclusions of this article is available in the NCBI Bioproject repository: Bioproject ID PRJNA531833, http://www.ncbi.nlm.nih.gov/bioproject/531833.

# Supplementary Information

# Acknowledgements

We would like to thank Keith Burling of the Core Biochemical Assay Laboratory, Cambridge, and Tertius Hough, Pathology Operations Manager at Medical Research Council Harwell Institute, for analysing the plasma and urine samples. We are also grateful to the Pathology Facility and the Light Microscopy Facility of the Barts Cancer Institute for their help with histology and imaging. All acknowledged authors have provided permission to be named in the manuscript. Sources of funding: This work was supported by Barts Charity grant MGU0361 to JL Williams, the People Programme (Marie Curie Actions) of the European Union's Seventh Framework Programme (FP7/2007–2013) under Rea grant agreement 608765 to S Awad, the Medical Research Council grant 165071 to J Nicholson, by a Marie Sklodowska Curie Innovative Training Network programme award (721532) to D Grzesik, a Medical Research Council grant MR/R02426X/1 to J Botta, a British Heart Foundation grant RG/15/15/31742 to A Tinker, and Medical Research Council grant MC_U142661184 to RD Cox. The work is facilitated by the National Institute for Health Research Barts Biomedical Research Centre.

### Author Contributions

JL Williams: conceptualization, resources, data curation, software, formal analysis, validation, investigation, visualization, methodology, project administration, and writing—original draft, review, and editing.
A Paudyal: resources and data curation.
S Awad: software, formal analysis, and visualization.
J Nicholson: software, formal analysis, and investigation.
D Grzesik: data curation.
J Botta: resources and methodology.
E Meimaridou: conceptualization, data curation, and investigation.
AV Maharaj: data curation.
M Stewart: conceptualization, data curation, formal analysis, and project administration.
A Tinker: resources, supervision, methodology, and writing—review and editing.

RD Cox: funding acquisition, investigation, methodology, and project administration.

LA Metherell: conceptualization, resources, formal analysis, supervision, funding acquisition, investigation, methodology, project administration, and writing—original draft, review, and editing.

## Conflict of Interest Statement

The authors declare that they have no conflict of interest.

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
