## [Reviewer comments · Life Science Alliance]

Life Science Alliance

Mylk3 null C57BL/6N mice develop cardiomyopathy whereas Nnt null C57BL/6J do not

Jack Williams, Anju Paudyal, Sherine Awad, James Nicholson, Dominika Grzesik, Joaquin Botta, Eirini Meimaridou, Avinaash Maharaj, Michelle Stewart, Andrew Tinker, Roger Cox, and Louise Metherell
DOI: <https://doi.org/10.26508/lsa.201900593>

Corresponding author(s): Louise Metherell, Queen Mary University Of London

Review Timeline:

Submission Date:	2019-10-30
Editorial Decision:	2019-11-17
Revision Received:	2020-02-17
Editorial Decision:	2020-03-03
Revision Received:	2020-03-10
Accepted:	2020-03-10

Scientific Editor: Andrea Leibfried

Transaction Report:

November 17, 2019

Re: Life Science Alliance manuscript #LSA-2019-00593-T

Louise Metherell
Centre for Endocrinology, William Harvey Research Institute, Barts and the London School of
Medicine and Dentistry, Queen Mary University of London
London EC1M 6BQ
United Kingdom

Dear Dr. Metherell,

Thank you for submitting your manuscript entitled "Mylk3 null C57BL/6N mice develop cardiomyopathy whereas Nnt null C57BL/6J do not" to Life Science Alliance. The manuscript was assessed by expert reviewers, whose comments are appended to this letter.

As you will see, your work received mixed views from the experts who evaluated it. Reviewer #3 thinks that your findings do not provide a significant value to others in the field in light of prior knowledge. Reviewer #1 and #2 are more positive, but think that more mouse work is required to better support your conclusions and to better link Mylk3 to the phenotype observed. We discussed your work in light of these conflicting views and decided to invite you to submit a revised version to us. The revised version should fully address all reviewers concerns.

Thank you for this interesting contribution to Life Science Alliance. We are looking forward to receiving your revised manuscript.

Sincerely,

B. MANUSCRIPT ORGANIZATION AND FORMATTING:

Reviewer #1 (Comments to the Authors (Required)):

In this manuscript, Williams et al. investigate the cardiac phenotype in two commonly used inbred mouse strains: C57BL/6J and C57BL/6N. It has been reported by others that C57BL/6J mice develop cardiomyopathy with age due to a null mutation in the Nnt gene encoding nicotinamide nucleotide

transhydrogenase. To test this idea the authors restored NNT expression in C57BL/6J mice via BAC transgenesis and study their heart function by echocardiography. Contrary to previous studies linking Nnt mutations to cardiovascular disease, neither the B6J or B6J; Nnt BAC strains showed signs of cardiac dysfunction.

On the other hand, the authors discovered that the B6N strain (control), not the B6J strain, developed cardiomyopathy with age (12 months old). Using molecular genetic analyses, the authors discovered a mutation in the Mylk3 gene in the B6N strain which abolishes MYLK3 protein expression. The authors propose that the Mylk3 mutation is responsible for the modest cardiomyopathy observed in the B6N strain.

Comments:

Understanding the phenotypic differences between these two commonly used mouse strains has important implications for designing better animal models of human disease. The Mylk3 mutation identified in the B6N strain may exacerbate the cardiac phenotype in mouse models of human heart disease, hence this information should be taken into consideration when deciding whether to use the B6N or B6J strain.

My primary concern is the temporal expressivity of this modest cardiomyopathy phenotype especially in older mice (18 months). While the LV dimensions and cardiac function measurements using echocardiography indicate cardiomyopathy at 12 months of age, fibrosis and significant changes in cardiac fetal gene expression normally associated with cardiomyopathy are absent even in older B6N mice (18 months of age). How do the authors explain the lack of disease progression in the B6N mice?

The manuscript as currently constructed is not focused on the primary findings (i.e., characterization of the B6N cardiac phenotype). The initial description of the study is misleading (p. 4) and doesn't fit with the title of the manuscript.

p. 4 'In this study we aimed to define the role of NNT in cardiomyopathy using the B6J NNT-null mice and a rescue model of the B6J mice, The authors did not observe a cardiac phenotype in the B6J mice; therefore, the B6J; Nnt BAC rescue experiment does not add any new information to the study and can be removed from the manuscript. Instead, to establish Mylk3 as the causative gene responsible for the cardiomyopathy in the B6N strain, the authors should rescue the cardiac phenotype in B6N mice with a BAC expressing the Mylk3 gene.

The authors need to carefully go through the figures and accurately indicate which strains/ages are being compared and whether there was a statistic difference between the groups. For example, the statistical significance of the differences between echo data among the groups is not shown in Suppl. Figure 1.

p. 10 'We also observed no increased fibrosis in B6N hearts at 18 months, barring one individual in the B6N group (Fig. 2b, c).' The authors only examined 5 animals per group, if the penetrance/expressivity of the phenotype is variable it may be necessary to examine additional animals before drawing any conclusions about the cardiac pathology in the older B6N mice. The authors mention one mouse with an unusual phenotype that appears to be an outlier. Since this mouse is not representative of the phenotype, it would be better to move the histology panel to supplemental data along with the other data from this unusual mouse.

Cardiac fibrosis is commonly observed with progressive cardiomyopathy, it is surprising 18 month old B6N animals did not exhibit fibrosis. Do the authors have any potential explanation?

The authors claim that MYLK4 upregulation may partially compensate for loss of MYLK3 in B6N hearts. The authors should check the phosphorylation status of the MYLK3 target, MYL2. Variable MYL2 phosphorylation among the B6N mice may explain the modest cardiomyopathy.

The finding of cardiomyopathy in the B6N strain rather than the B6J strain is unexpected. The authors should discuss their results in the context of previous studies examining cardiac function between these two strains.

General comments:

Fig. 1a: It is difficult to compare the echo data. Typically, only the echo traces are shown not a screen shot. Better to show higher magnification image of the traces.

Fig. 1o: Heart weights are typically shown relative to tibia length or body weight. The authors mention tibia length in the text, but it seems the heart weight/tibia length was not calculated for each individual mouse as only heart weight is included in Fig. 1o. The authors need to clarify this point.

Fig. 2a: To illustrate modest differences in chamber dimensions, heart sections in the short axis are preferred to the long axis view shown.

Fig. 4a,b,c: Why show the Nnt gene expression data? It is not relevant to interpreting the B6N cardiac phenotype which is the focus of the paper.

Fig. 5e,f,g: It is not necessary to show western data of MYLK3 expression at 3 time points, 3 month time point is sufficient.

Fig. 5d: To demonstrate that MYLK3 is not expressed in B6N hearts, western analysis is sufficient. Immunostaining for the purpose of protein quantification is not necessary.

In figure legends, asterisks ** and *** both indicate a p value < 0.01.

Reviewer #2 (Comments to the Authors (Required)):

In this interesting and important study Williams investigated cardiac performance and morphology in three different C57BL/6 sub-strains at 3, 12, and 18 months of age. C57BL/6 mice and in particular C57BL/6N are commonly used as genetic background in genetic loss-of-function studies. C57BL/6J mice harbor a mutation in the Nnt gene (nicotinamide nucleotide transhydrogenase gene), associated with familial glucocorticoid deficiency but not with a heart phenotype in humans. Hence, the authors also employed a transgenic C57BL/6J strain that carries a BAC containing the Nnt gene, which should reverse any potential phenotype in WT C57BL/6J mice. Surprisingly, the authors detected a moderate form of dilated cardiomyopathy in C57BL/6N but not in C57BL/6J mice. Cardiomyocytes of C57BL/6N mice showed eccentric hypertrophy but preserved sarcomeres. RNA-seq revealed increased numbers of differentially expressed genes in C57BL/6N hearts, which together with variant calling led to identification of a mutation in the Mylk3 gene. The mutation introduces a putative translation initiation site (TIS), 5bp upstream of the canonical TIS disrupting the open reading frame and essentially abolishing Mylk3 protein expression. The authors speculate that upregulation of Mylk4, which is primarily expressed at embryonic and fetal stages,

compensates to a large degree for loss of Mylk3 function, but fails to fully prevent altered phosphorylation of the myosin regulatory light chain (MYL2), eventually leading to a cardiac phenotype.

The study is well executed and emphasizes the need for a comprehensive characterization of genetic backgrounds used for genetic studies. An important insight from the current study is that we have to investigate potential synthetic effects when introducing mutations putatively affecting cardiac functions in C57BL/6N mice, which already carry a defect in the Mylk3 gene. Although the topic is highly relevant for biomedical research and the study is well-designed and up-to-date including expert evaluation of the bio-statistical data, I noticed a few shortcomings. Apparently, the authors started out to detect a phenotype in C57BL/6J mice, which is probably the reason why they included C57BL/6J mice carrying a Nnt-BAC in the study. Since no noteworthy cardiac phenotype was detected in C57BL/6J mice, the use of C57BL/6J-BAC-Nnt is somewhat obsolete. Instead, the authors should have used C57BL/6N mice carrying the Mylk3 on a BAC to unequivocally demonstrate that the mutation in the Mylk3 gene is responsible for the cardiomyopathy.

Specific comments

A direct proof that the mutation in the Mylk3 gene detected in the genome of C57BL/6N mice is responsible for the cardiomyopathy is missing. I do not deny that the correlations are compelling, including the up-regulation of the Mylk4 gene. However, such correlations do not qualify as proof. In principle, other not yet detected mutations in the C57BL/6N genome might account for the phenotype in C57BL/6N mice. The study would be much more compelling when the authors would use C57BL/6N mice carrying a Mylk3-BAC instead of C57BL/6J mice carrying a Nnt-BAC, which anyway does not make much sense in the current setting. Alternatively, the authors might compare C57BL/6J mice to Mylk3 mutants, which have been recently described to develop a mild type of cardiomyopathy (<https://doi.org/10.3389/fphys.2019.00696>).

It might be very helpful to investigate the phosphorylation of MYL2 in C57BL/6N mice at different ages and compare to C57BL/6J animals. Is there any difference in the extent of MYL2 phosphorylation between the different ages and strains?

The moderate ventricular dilatation loss and reduction of cardiac function in C57BL/6N mice is interesting. It is important to know whether the phenotype is clinically relevant, i.e. are there any relevant differences regarding long-term survival?

I do not really understand why the authors devoted a paragraph and two supplemental figures in the manuscript to the analysis of putative transcription factors that might control deregulated genes. The analysis yielded no insights and is clearly dispensable.

Minor points

In Fig. 2, the authors show images of 'stretched' cardiomyocytes from C57BL/6N mice but not from C57BL/6J and C57BL/6J BAC mice, but cardiomyocytes from all three groups were quantified. In contrast, images of all three groups are shown in (g). To maintain consistency, images of cardiomyocytes from C57BL/6J and C57BL/6J BAC mutants should be shown as well.

The staining with phalloidin and for Tnnt2 in Fig.2g looks very similar and thus unspecific. Phalloidin stains for F-actin, which shows a different pattern than Tnnt2. Please replace the images with something more suitable.

According to the images, ventricles of C57BL/6N barely show any Tnnt2 staining, whereas those

from C57BL/6J BAC are highly positive. In the text, the authors state that there are no differences (which is probably correct but not reflected by the images). Again, more suitable images should be used for the comparisons. Proper quantifications should be done when expression levels are compared.

Error bars are missing in Fig 6a/b.

Reviewer #3 (Comments to the Authors (Required)):

Summary:

Authors aimed to define the role of NNT in cardiomyopathy using B6 subtypes J and N, where its expression varies. After having concluded that NNT expression is not associated with cardiomyopathy, they characterized the differences between J and N, by echocardiography of mice at 3, 12, 18 months of age, and some comparisons of myocyte morphology and RNAseq of old 18 month old mice. A variant in the MYLK3, that is unique to N subtype was identified, and validated by luciferase assay. Expression of the protein encoded by MYLK3 was confirmed by immunoblotting. The conclusions are, that they 1) described a Mylk3 mutation that is responsible for a modest cardiomyopathy in the C57BL/6N substrain and demonstrated that NNT ablation in the C57BL/6J mouse does not, by itself, cause a cardiac phenotype, 2) demonstrated how a single gene defect can cause susceptibility to a particular phenotype in a mouse substrain, in this case cardiomyopathy in the B6N. Thus, they feel this study accentuates the importance, in a wider setting, of knowing the genetic defects in your mouse substrain and highlights the value of selecting your mouse background with care.

Major comments:

For conclusion 1, evidence that substantiates this claim is weak in that the mutation was already reported 4 years ago, and differences in expression of cMLCK in the hearts of N and J substrains were already described, and well-known in the cardiac muscle field. Additionally, substrain differences between N and J are well-documented in various other research fields.

For conclusion 2, they did demonstrate that a single gene defect can cause a phenotype, by comparing two substrains, but this is an obvious result, and already described by other groups.

Areas to address:

- 1) western blot for MYLK4 should be included, and this should not take more than 1 week, provided that samples are still available.
- 2) discussion on substrain differences in RLC phosphorylation levels should be included. MYLK4 RNA expression is higher in N substrain but previously published RLC phosphorylation is lower in these mice, which other authors have shown is the cause for cardiac dysfunction in the N substrain mice.
- 3) manuscript needs to be proof-read for typographical errors.
 - a. Page 4-misplaced "j" after refs 18,19
- 4) references need to be checked in various places.
 - a. page 5, references 25 and 26 does not support the sentence.
 - b. page 13- Mylk3 is not highly expressed in skeletal muscle, and the references cited does not address skeletal muscle expression.

This manuscript summarizes a body of work that is descriptive and does not directly address or answer a scientific hypothesis driven question. As many of the points made were already published by others, this manuscript may not receive much attention from the scientific community. This reviewer feels that the authors missed the opportunity to highlight the true novelty of this study: analysis of old mice substrains and susceptibility to overt phenotype in responses to systemic stressors, like inflammation, which would elicit increased interest by a broader audience.

Reviewer #1 (Comments to the Authors (Required)):

In this manuscript, Williams et al. investigate the cardiac phenotype in two commonly used inbred mouse strains: C57BL/6J and C57BL/6N. It has been reported by others that C57BL/6J mice develop cardiomyopathy with age due to a null mutation in the *Nnt* gene encoding nicotinamide nucleotide transhydrogenase. To test this idea the authors restored NNT expression in C57BL/6J mice via BAC transgenesis and study their heart function by echocardiography. Contrary to previous studies linking *Nnt* mutations to cardiovascular disease, neither the B6J or B6J; *Nnt* BAC strains showed signs of cardiac dysfunction.

On the other hand, the authors discovered that the B6N strain (control), not the B6J strain, developed cardiomyopathy with age (12 months old). Using molecular genetic analyses, the authors discovered a mutation in the *Mylk3* gene in the B6N strain which abolishes MYLK3 protein expression. The authors propose that the *Mylk3* mutation is responsible for the modest cardiomyopathy observed in the B6N strain.

Comments:

Understanding the phenotypic differences between these two commonly used mouse strains has important implications for designing better animal models of human disease. The *Mylk3* mutation identified in the B6N strain may exacerbate the cardiac phenotype in mouse models of human heart disease, hence this information should be taken into consideration when deciding whether to use the B6N or B6J strain.

My primary concern is the temporal expressivity of this modest cardiomyopathy phenotype especially in older mice (18 months). While the LV dimensions and cardiac function measurements using echocardiography indicate cardiomyopathy at 12 months of age, fibrosis and significant changes in cardiac fetal gene expression normally associated with cardiomyopathy are absent even in older B6N mice (18 months of age). How do the authors explain the lack of disease progression in the B6N mice?

It has previously been shown that fibrosis and fetal gene expression is induced in the B6N but not the B6J mice, but only following a course of transverse aortic banding or Angiotensin II treatment (Cardin et al., 2014, Garcia-Menendez et al., 2014). In our study we see no evidence of increased fibrosis in the B6N, however, there is some evidence of increased foetal gene expression which would fit with cardiac remodelling. So while our mice do not naturally develop the fibrosis typical of cardiomyopathy, we believe this can, and has been induced by a second trigger in previous studies whereby the B6N develop a far more severe phenotype than the B6J. That trigger may be TAC or Angiotensin, or may be a second gene knockout/modifier.

The manuscript as currently constructed is not focused on the primary findings (i.e., characterization of the B6N cardiac phenotype). The initial description of the study is misleading (p. 4) and doesn't fit with the title of the manuscript.

p. 4 'In this study we aimed to define the role of NNT in cardiomyopathy using the B6J NNT-null mice and a rescue model of the B6J mice, The authors did not observe a cardiac phenotype in the B6J mice; therefore, the B6J; *Nnt* BAC rescue experiment does not add any new information to the study and can be removed from the manuscript. Instead, to establish *Mylk3* as the causative gene responsible for the cardiomyopathy in the B6N strain, the

authors should rescue the cardiac phenotype in B6N mice with a BAC expressing the Mylk3 gene.

We are reporting two findings in this paper, not only the cardiomyopathy that we discovered in the B6N substrain but also the proof that NNT loss, at least in the mouse, does not cause cardiomyopathy. This could not have been elucidated without the B6J-BAC. Removing the BAC mouse from the study would therefore be a mistake. The reviewer is correct to suggest we rescue the phenotype in the B6N background, in collaboration with MRC Harwell Institute, we are currently generating a Crispr-targeted knock-in of the allele from the B6J, rather than a BAC. However, the completion of this study is optimistically two years away and we believe the results warrant publication now, especially in light of the ongoing studies of cardiac phenotypes in the B6N by the International Mouse Phenotyping Consortium (IMPC) and other researchers, allowing them to choose the optimal substrain for their research.

The authors need to carefully go through the figures and accurately indicate which strains/ages are being compared and whether there was a statistical difference between the groups. For example, the statistical significance of the differences between echo data among the groups is not shown in Suppl. Figure 1.

The differences at each age group are shown in Fig1 (12 months) Fig S2 (3 months) and Fig S3 (18 months). We produced Fig S1 to show all the data collated, so the reader can more easily see the progression of each mouse strain across the age groups, rather than to display significance values. Trying to indicate significance scoring on this figure is quite messy due to the number of comparisons performed. This figure can be removed and we are happy to be guided by the editor on this point.

p. 10 'We also observed no increased fibrosis in B6N hearts at 18 months, barring one individual in the B6N group (Fig. 2b, c).' The authors only examined 5 animals per group, if the penetrance/expressivity of the phenotype is variable it may be necessary to examine additional animals before drawing any conclusions about the cardiac pathology in the older B6N mice. The authors mention one mouse with an unusual phenotype that appears to be an outlier. Since this mouse is not representative of the phenotype, it would be better to move the histology panel to supplemental data along with the other data from this unusual mouse. Cardiac fibrosis is commonly observed with progressive cardiomyopathy, it is surprising 18 month old B6N animals did not exhibit fibrosis. Do the authors have any potential explanation?

We believe the numbers used in this study are sufficient to conclude there is not a consistent fibrotic phenotype in our 6N mice. Alongside the five mice per group examined here, we have also examined the RNA expression of various pathways in our RNA-seq dataset in a separate cohort of 18-month old mice (n=5 per group) and find no difference in fibrotic genes (Table S4). We have moved the unusual data point into the supplementary, and instead show representative images from each genotype (Fig 2c). In regards to the lack of fibrosis, I refer the reviewer to our response to their first comment.

The authors claim that MYLK4 upregulation may partially compensate for loss of MYLK3 in B6N hearts. The authors should check the phosphorylation status of the MYLK3 target, MYL2. Variable MYL2 phosphorylation among the B6N mice may explain the modest cardiomyopathy.

As this reviewer states, variable MYL2 phosphorylation would be a plausible explanation for the modest cardiomyopathy. Unfortunately, there is no commercially available phospho-MYL2 antibody. The commonly used pMLC antibody that recognises serine 19 is not specific for MYL2 and does not target the residues phosphorylated by MYLK3 (S14, S15). We have tried to source antibodies from other groups who have published on this, e.g. Neal Epstein M.D., Hideko Kasahara, Danuta Szczesna-Cordary, but without success

The finding of cardiomyopathy in the B6N strain rather than the B6J strain is unexpected. The authors should discuss their results in the context of previous studies examining cardiac function between these two strains.

In both the introduction and discussion we have covered the (limited) literature regarding previous studies specifically examining cardiac function between these two substrains. We do not think we have missed any previous studies. There are many gene knockout studies using one or other mouse substrain and, on occasion, both. In many publications the specific C57BL/6 substrain used is not reported, and in some the knockout is produced on one background whilst the control used is on the other.

General comments:

Fig. 1a: It is difficult to compare the echo data. Typically, only the echo traces are shown not a screen shot. Better to show higher magnification image of the traces.

We have amended this in the upper panel of Fig1.

Fig. 1o: Heart weights are typically shown relative to tibia length or body weight. The authors mention tibia length in the text, but it seems the heart weight/tibia length was not calculated for each individual mouse as only heart weight is included in Fig. 1o. The authors need to clarify this point.

Regrettably we did not take tibia lengths from this cohort of mice, however we have subsequently performed this analysis in a separate cohort of 9 month old B6N and B6J mice and find no difference in heart weight/tibia length ratio.

Fig. 2a: To illustrate modest differences in chamber dimensions, heart sections in the short axis are preferred to the long axis view shown.

While we appreciate this is the case, this figure is not designed to show differences in chamber dimensions, rather to show the total size of the hearts is comparable, and there has not been any gross hypertrophy in the B6N mice, in line with the heart weight data. We have changed the text (line 225) and the figure to reflect this conclusion.

Fig. 4a,b,c: Why show the Nnt gene expression data? It is not relevant to interpreting the B6N cardiac phenotype which is the focus of the paper.

The reviewer is correct, this is irrelevant here and has been removed.

Fig. 5e,f,g: It is not necessary to show western data of MYLK3 expression at 3 time points, 3 month time point is sufficient.

We have removed two of the three time points in the main text, as suggested by this reviewer, moved the 12 and 18-month data to supplementary figures and updated the text to reflect this.

Fig. 5d: To demonstrate that MYLK3 is not expressed in B6N hearts, western analysis is sufficient. Immunostaining for the purpose of protein quantification is not necessary.

The images have been moved to supplementary data (Fig S5a,b) since they also serve to illustrate the spatial expression of MYLK3, which we have now highlighted in the text (line 325).

In figure legends, asterisks ** and *** both indicate a p value < 0.01.

This has been corrected.

Reviewer #2 (Comments to the Authors (Required)):

In this interesting and important study Williams investigated cardiac performance and morphology in three different C57BL/6 sub-strains at 3, 12, and 18 months of age. C57BL/6 mice and in particular C57BL/6N are commonly used as genetic background in genetic loss-of-function studies. C57BL/6J mice harbor a mutation in the Nnt gene (nicotinamide nucleotide transhydrogenase gene), associated with familial glucocorticoid deficiency but not with a heart phenotype in humans. Hence, the authors also employed a transgenic C57BL/6J strain that carries a BAC containing the Nnt gene, which should reverse any potential phenotype in WT C57BL/6J mice. Surprisingly, the authors detected a moderate form of dilated cardiomyopathy in C57BL/6N but not in C57BL/6J mice. Cardiomyocytes of C57BL/6N mice showed eccentric hypertrophy but preserved sarcomeres. RNA-seq revealed increased numbers of differentially expressed genes in C57BL/6N hearts, which together with variant calling led to identification of a mutation in the Mylk3 gene. The mutation introduces a putative translation initiation site (TIS), 5bp upstream of the canonical TIS disrupting the open reading frame and essentially abolishing Mylk3 protein expression. The authors speculate that upregulation of Mylk4, which is primarily expressed at embryonic and fetal stages, compensates to a large degree for loss of Mylk3 function, but fails to fully prevent altered phosphorylation of the myosin regulatory light chain (MYL2), eventually leading to a cardiac phenotype.

The study is well executed and emphasizes the need for a comprehensive characterization of genetic backgrounds used for genetic studies. An important insight from the current study is that we have to investigate potential synthetic effects when introducing mutations putatively affecting cardiac functions in C57BL/6N mice, which already carry a defect in the Mylk3 gene. Although the topic is highly relevant for biomedical research and the study is well-designed and up-to-date including expert evaluation of the bio-statistical data, I noticed a few shortcomings. Apparently, the authors started out to detect a phenotype in C57BL/6J mice, which is probably the reason why they included C57BL/6J mice carrying a Nnt-BAC in the study. Since no noteworthy cardiac phenotype was detected C57BL/6J mice, the use of C57BL/6J-BAC-Nnt is somewhat obsolete. Instead, the authors should have used C57BL/6N mice carry the Mylk3 on a BAC to unequivocally demonstrate that the mutation in the Mylk3 gene is responsible for the cardiomyopathy.

Specific comments

A direct proof that the mutation in the Mylk3 gene detected in the genome of C57BL/6N

mice is responsible for the cardiomyopathy is missing. I do not deny that the correlations are compelling, including the up-regulation of the *Myk4* gene. However, such correlations do not qualify as proof. In principle, other not yet detected mutations in the C57BL/6N genome might account for the phenotype in C57BL/6N mice. The study would be much more compelling when the authors would use C57BL/6N mice carrying a *Myk3*-BAC instead of C57BL/6J mice carrying a *Nnt*-BAC, which anyway does not make much sense in the current setting. Alternatively, the authors might compare C57BL/6J mice to *Myk3* mutants, which have been recently described to develop a mild type of cardiomyopathy (<https://doi.org/10.3389/fphys.2019.00696>).

While the reviewer is correct to point this out, we set out to examine the effect of *NNT* expression on cardiomyopathy, since *NNT* variants have been linked to cardiomyopathy in humans and the design of the study and the mouse substrains utilised reflects this. It was only with the inclusion of the B6J BAC that the difference in cardiac phenotype was attributed to substrain difference. In collaboration with MRC Harwell Institute we are currently generating a rescue mouse by a Crispr-targeted knock-in of the *Myk3* variant from the B6J, rather than a BAC. However, the results of this study are two years away and we believe the results warrant publication now, especially in light of the ongoing studies of cardiac phenotypes in the B6N by the International Mouse Phenotyping Consortium and other researchers, allowing them to choose the optimal substrain for their research.

It might be very helpful to investigate the phosphorylation of MYL2 in C57BL/6N mice at different ages and compare to C57BL/6J animals. Is there any difference in the extent of MYL2 phosphorylation between the different ages and strains?

As this reviewer and reviewer #1 state, variable MYL2 phosphorylation would be a plausible explanation for the modest cardiomyopathy. Unfortunately, there is no commercially available phospho-MYL2 antibody. The commonly used pMLC antibody that recognises serine 19 is not specific for MYL2. In an attempt to circumvent this issue we performed an immunoprecipitation experiment with MYL2 and then blotted with a pan-phosphoserine antibody, finding no difference in phosphorylation. However we do not believe the results from this are interpretable either as there are multiple other phosphoserine sites beside the reported S14 and S15 sites phosphorylated by MYLK3. We have tried to source antibodies from other groups who have published on this, e.g. Neal Epstein M.D., Hideko Kasahara, Danuta Szczesna-Cordary, but without success.

The moderate ventricular dilatation loss and reduction of cardiac function in C57BL/6N mice is interesting. It is important to know whether the phenotype clinically relevant, i.e. are there any relevant differences regarding long-term survival?

We do not notice any differences in survival, which we have noted in the text (line 133, line 280). We believe the phenotype is subclinical. Rather the B6N is 'primed' and has a far worse survival rate than the B6J, following a challenge such as transverse aortic constriction^{1,2}, as has been shown in other studies which we have duly referred to in the text (line 307 to 315).

I do not really understand why the authors devoted a paragraph and two supplemental figures

in the manuscript to the analysis of putative transcription factors that might control deregulated genes. The analysis yielded no insights and is clearly dispensable.

We agree with the reviewer and have removed this section.

Minor points

In Fig. 2, the authors show images of 'stretched' cardiomyocytes from C57BL/6N mice but not from C57BL/6J and C57BL/6J BAC mice, but cardiomyocytes from all three groups were quantified. In contrast, images of all three groups are shown in (g). To maintain consistency, images of cardiomyocytes from C57BL/6J and C57BL/6J BAC mutants should be shown as well.

This has been corrected (Fig 2e, f).

The staining with phalloidin and for Tnnt2 in Fig.2g looks very similar and thus unspecific. Phalloidin stains for F-actin, which shows a different pattern than Tnnt2. Please replace the images with something more suitable.

Thank you for this guidance. We have since performed staining with better markers of sarcomere assembly, myosin light chain 2 and alpha actinin, and the analysis has been repeated, we have replaced the original images with these results. Surprisingly, we now find a difference in sarcomere structure (Fig 2i-l), which is discussed in the results (line 155) and the discussion (line 268).

According to the images, ventricles of C57BL/6N barely show any Tnnt2 staining, whereas those from C57BL/6J BAC are highly positive. In the text, the authors state that there are no differences (which is probably correct but not reflected by the images). Again, more suitable images should be used for the comparisons. Proper quantifications should be done when expression levels are compared.

We have replaced the images as mentioned above, and the western blot in the next panel confirms no difference in protein expression of Tnnt2. These blots were quantified by densitometry, but no statistical difference was found (line 160)

Error bars are missing in Fig 6a/b.

Error bars have been added.

Reviewer #3 (Comments to the Authors (Required)):

Summary:

Authors aimed to define the role of NNT in cardiomyopathy using B6 subtypes J and N, where its expression varies. After having concluded that NNT expression is not associated with cardiomyopathy, they characterized the differences between J and N, by echocardiography of mice at 3, 12, 18 months of age, and some comparisons of myocyte morphology and RNAseq of old 18 month old mice. A variant in the MYLK3, that is unique to N subtype was identified, and validated by luciferase assay. Expression of the protein

encoded by MYLK3 was confirmed by immunoblotting. The conclusions are, that they 1) described a Mylk3 mutation that is responsible for a modest cardiomyopathy in the C57BL/6N substrain and demonstrated that NNT ablation in the C57BL/6J mouse does not, by itself, cause a cardiac phenotype, 2) demonstrated how a single gene defect can cause susceptibility to a particular phenotype in a mouse substrain, in this case cardiomyopathy in the B6N. Thus, they feel this study accentuates the importance, in a wider setting, of knowing the genetic defects in your mouse substrain and highlights the value of selecting your mouse background with care.

Major comments:

For conclusion 1, evidence that substantiates this claim is weak in that the mutation was already reported 4 years ago, and differences in expression of cMLCK in the hearts of N and J substrains were already described, and well-known in the cardiac muscle field. Additionally, substrain differences between N and J are well-documented in various other research fields.

While the reviewer is correct to note the previous publication, we clearly referenced this in our manuscript. We are the first to describe a DCM phenotype in these mice, perhaps because we aged them out beyond other studies, and also the first to link the gene variant with the cardiac phenotype. We do not agree that these effects are well known. This is reflected in the study we refer to in our conclusions where over 60% of references to the C57BL/6 mouse in the journal *Diabetes* make no mention of the particular substrain. We also note in the manuscript introduction the publication of other data enumerating differences in other phenotypes between the strains, but nobody has ever identified a DCM phenotype in these mice and linked it to the variant described. We have performed a comprehensive phenotyping which clearly and undeniably demonstrates a DCM phenotype in a mouse used widely all over the world, indeed the very model used by the IMPC to generate its knockout library.

For conclusion 2, they did demonstrate that a single gene defect can cause a phenotype, by comparing two substrains, but this is an obvious result, and already described by other groups.

The specific conclusion that the *Mylk3* variant in the B6N substrain leads to a modest dilated cardiomyopathy in adult mice has not been described by any other group. While the variant was previously published, it was not linked to a differential phenotype in the hearts, only in isolated papillary muscles, which we have referred to in our text.

Areas to address:

1) western blot for MYLK4 should be included, and this should not take more than 1 week, provided that samples are still available.

This has been performed and showed no difference in MYLK4 expression. We have updated our conclusions to reflect this (line 254 in results, line 301 in discussion).

2) discussion on substrain differences in RLC phosphorylation levels should be included. MYLK4 RNA expression is higher in N substrain but previously published RLC phosphorylation is lower in these mice, which other authors have shown is the cause for cardiac dysfunction in the N substrain mice.

With reference to the discussion, we believe we have sufficiently covered what is known regarding MYL2 phosphorylation between the substrains, as there is only one paper that has published on this topic. As the reviewer mentions, this paper showed slightly reduced MYL2 phosphorylation, though one might argue that this was not proved to be the cause for the cardiac dysfunction in these mice.

3) manuscript needs to be proof-read for typographical errors.

a. Page 4-misplaced "j" after refs 18,19

4) references need to be checked in various places.

a. page 5, references 25 and 26 does not support the sentence.

b. page 13- Mylk3 is not highly expressed in skeletal muscle, and the references cited does not address skeletal muscle expression.

These have been corrected.

This manuscript summarizes a body of work that is descriptive and does not directly address or answer a scientific hypothesis driven question. As many of the points made were already published by others, this manuscript may not receive much attention from the scientific community. This reviewer feels that the authors missed the opportunity to highlight the true novelty of this study: analysis of old mice substrains and susceptibility to overt phenotype in responses to systemic stressors, like inflammation, which would elicit increased interest by a broader audience.

The original hypothesis, that NNT loss seen in B6J mice causes cardiomyopathy, is explicitly stated not only at the end of the introduction but also at the start of the discussion. In disproving this hypothesis we discovered a modest cardiomyopathy in the B6N mice and went on to link this to the gene defect in *Mylk3*. Although the differences in cardiac phenotype and the B6N *Mylk3* variant have previously been described we are the first to demonstrate the link between the two. Regarding the response of these mice to stressors, we have not subjected our mouse cohorts to stressors so cannot draw any conclusion about their possible effect(s) on ageing mice. We believe the substrain differences to stressors have already been covered by other groups (Cardin et al., 2014, Garcia-Menendez et al., 2014). We feel the paper will have broad interest since the B6N and B6J mouse substrains are the most commonly used for studies worldwide.

March 3, 2020

RE: Life Science Alliance Manuscript #LSA-2019-00593-TRR

Prof. Louise Metherell
Queen Mary University Of London
Centre for Endocrinology,
Barts and the London School of Medicine and Dentistry, Queen Mary University of London
London EC1M 6BQ
United Kingdom

Dear Dr. Metherell,

Thank you for submitting your revised manuscript entitled "Myk3 null C57BL/6N mice develop cardiomyopathy whereas Nnt null C57BL/6J do not". Two of the original reviewers re-evaluated your study and you can find their comments below. Based on the input received, we would be happy to publish your paper in Life Science Alliance, pending final revisions:

- It is important to address the remaining reviewer concerns carefully, please do so
- Please make sure that the author names listed on your manuscript file and within our system match (eg, Andy/Andrew Tinker)
- Please make sure that figure legends can stand on their own (ie, please render S figure legends more informative)
- Please upload all figures with the correct figure labeling / name (eg., current figure S1 is not showing data described for S1)
- Please either mention all individual panels of a certain figure in your manuscript text or only the figure itself (eg, there is a callout for Fig S1o, but not for S1a-n; there is a callout to Fig S4b, but not S4a)
- Please add callouts in the manuscript text to figures S2f-n, S3j, m, n
- Please provide tables S1, S3, S4 in excel or docx format
- Fig. 6e is missing a callout in the manuscript text and description in figure legend
- Sub-panels in the figure S4 seem random, please fix
- The data availability could not get checked (BioProject 531833 not public), please make the data available and make sure to deposit the RNA-seq data to a repository
- Please include a statement in your manuscript to confirm that all mouse experiments were performed in accordance with relevant guidelines and regulations
- The statement "Primer sequences are available in the online supplement" misses the respective primer data

A. FINAL FILES:

B. MANUSCRIPT ORGANIZATION AND FORMATTING:

Sincerely,

Reviewer #1 (Comments to the Authors (Required)):

The authors have addressed most of my concerns in the revised manuscript. However, a few statements need to be corrected/revised before publication.

The authors repeatedly overstate their findings in the paper. Based on what we know from human and mouse models, I agree that the Mylk3 mutation is likely responsible for the cardiomyopathy in the B6N mice. However, as the authors recognize, to prove this point the B6N heart phenotype will need to be rescued by introduction of the wild-type Mylk3 gene. The following statements need to be modified to reflect the actual findings presented by the authors.

Abstract

51 we identified a null mutation in Mylk3 driving a cardiomyopathy phenotype in the C57BL/6N.

The C57BL/6N mouse may therefore provide the first clinically relevant DCM model for therapeutic drug testing. THE B6N MOUSE IS NOT THE FIRST CLINICALLY RELEVANT DCM MOUSE MODEL. There are many mouse models of DCM, e.g., TNNT2 KI mutant mouse model (Du et al., 2007).

Introduction

95 We go on to show that the CM in B6N is due to a null mutation in myosin light chain kinase 3 (Mylk3) which is only present in B6N substrains.

Results

266 Instead, we identify a mutation in Mylk3 in B6N mice that abolishes protein expression and causes dilated cardiomyopathy.

Discussion

367 We have described a Mylk3 mutation that is responsible for a modest cardiomyopathy in the C57BL/6N substrain

The authors need to include the significance values for the echo data in Suppl Table I.

To make it easier for the reader to follow the study and interpret the data, the authors should use more descriptive terms.

To ascertain whether the B6J substrain developed cardiomyopathy, male mice from the

104 three substrains B6N, B6J and B6J-BAC ??? were analysed at 3, 12 and 18 months of age,

The authors should refer to the B6J rescue mouse as B6J-Nnt instead of B6J-BAC throughout the paper.

241 Luciferase activity was significantly lower in HEK293T cells transfected with the 'B6N-Luc' ??? vector compared to those transfected with 'B6J-Luc', indicating the mutation is sufficient to abrogate protein translation (Fig 6b).

To help the reader, the authors should specify the luciferase constructs, i.e., refer to B6N-Luc as B6N-5UTR-MYLK3 and B6J-Luc as B6J-5UTR-MYLK3. Or simply B6J-5UTR

Reviewer #2 (Comments to the Authors (Required)):

Williams et al have submitted a revised version of their manuscript, which describes the detection of a moderate form of dilated cardiomyopathy in C57BL/6N but not in C57BL/6J mice, which might be due to a mutation in the Mylk3 gene, probably also leading to increased expression of Mylk4. The authors improved the manuscript by addressing several comments raised by the reviewers. Regrettably, the most critical problems were not dealt with. In my original review I pointed out that a direct proof that the mutation in the Mylk3 gene detected in the genome of C57BL/6N mice is responsible for the cardiomyopathy is missing. The authors responded that they are aware of this problem but that Crispr-targeted knock-in of the Mylk3 variant is underway but that it will take two years to obtain and analyse the mice. In fact, I always have an uneasy feeling when recommending additional mouse experiments (or even generation of new strains), since I am well aware how much time and efforts this requires. I agree with the authors that the publication requires timely publication to make the scientific community aware of the problems with the C57BL/6N mouse strain. However, one needs to be aware that this strategy reduces the scientific relevance of the current study.

Unfortunately, the phosphorylation analysis did not generate useful information, although the authors seemed to tried hard. There is probably not much more that can be done in this direction at the moment.

All other issues have been dealt with appropriately. In particular, the quality of the images was improved, which also lead to the detection of changes in the sarcomere structure.

Dear Dr. Liebfried,

We are pleased to submit our corrected version of our paper entitled '***Mylk3 null C57BL/6N mice develop cardiomyopathy whereas Nnt null C57BL/6J do not***'.

We would like to thank yourself and the reviewers for your suggestions. Please see below a list of the requested changes from yourself and the reviewers and our point-by-point explanation of how we have addressed each change.

Final Revisions as requested by the Editor:

- Please make sure that the author names listed on your manuscript file and within our system match (eg, Andy/Andrew Tinker). **This has been amended in the latest version.**
- Please make sure that figure legends can stand on their own (ie, please render S figure legends more informative). **This has been amended. See figure S1, S2, and S3.**
- Please upload all figures with the correct figure labeling / name (eg., current figure S1 is not showing data described for S1). **This appears to have been an uploading issue and has been amended.**
- Please either mention all individual panels of a certain figure in your manuscript text or only the figure itself (eg, there is a callout for Fig S1o, but not for S1a-n; there is a callout to Fig S4b, but not S4a). **We have altered the text to include callouts for all data, see line 126 and line 167**
- Please add callouts in the manuscript text to figures S2f-n, S3j, m, n. **We have altered the text to include callouts to this data, see line 121 and line 123.**
- Please provide tables S1, S3, S4 in excel or docx format. **These files are now in Excel format.**
- Fig. 6e is missing a callout in the manuscript text and description in figure legend. **There is no Figure 6e, the figure runs to 'd' only. We have checked whether this was a typo and applied to another figure, but find no such evidence other than those described in the points above.**
- Sub-panels in the figure S4 seem random, please fix. **We have included more**

information in the figure legend to indicate the sub-panel classification, which we believe clarifies the choice of sub-panel.

- The data availability could not get checked (BioProject 531833 not public), please make the data available and make sure to deposit the RNA-seq data to a repository. The RNAseq data has been deposited under reference Bioproject ID PRJNA531833, <http://www.ncbi.nlm.nih.gov/bioproject/531833>. The data is embargoed until publication date but will then be freely available.

- Please include a statement in your manuscript to confirm that all mouse experiments were performed in accordance with relevant guidelines and regulations. This has been added, line 384.

- The statement "Primer sequences are available in the online supplement" misses the respective primer data. This has been amended, the online supplement is uploaded in the latest version.

Reviewer #1 (Comments to the Authors (Required)):

The authors have addressed most of my concerns in the revised manuscript. However, a few statements need to be corrected/revised before publication.

The authors repeatedly overstate their findings in the paper. Based on what we know from human and mouse models, I agree that the Mylk3 mutation is likely responsible for the cardiomyopathy in the B6N mice. However, as the authors recognize, to prove this point the B6N heart phenotype will need to be rescued by introduction of the wild-type Mylk3 gene. The following statements need to be modified to reflect the actual findings presented by the authors.

Abstract

51 we identified a null mutation in Mylk3 driving a cardiomyopathy phenotype in the C57BL/6N.

This has been amended to:

51 we identified a null mutation in Mylk3 as a credible cause of the cardiomyopathy phenotype in the C57BL/6N.

The C57BL/6N mouse may therefore provide the first clinically relevant DCM model for therapeutic drug testing. THE B6N MOUSE IS NOT THE FIRST CLINICALLY RELEVANT DCM MOUSE MODEL. There are many mouse models of DCM, e.g., TNNT2 KI mutant mouse model (Du et al., 2007). This statement has been deleted.

Introduction

95 We go on to show that the CM in B6N is due to a null mutation in myosin light chain kinase 3 (Mylk3) which is only present in B6N substrains.

This has been amended to:

95 We go on to show that the CM in B6N may be due to a null mutation in myosin light chain kinase 3 (Mylk3) which is only present in B6N substrains.

Results

266 Instead, we identify a mutation in Mylk3 in B6N mice that abolishes protein

expression and causes dilated cardiomyopathy.

This has been amended to:

266 Instead, we identify a mutation in Mylk3 in B6N mice that abolishes protein expression and **likely** causes dilated cardiomyopathy.

Discussion

367 We have described a Mylk3 mutation that is responsible for a modest cardiomyopathy in the C57BL/6N substrain.

This has been amended to:

367 We have described a Mylk3 mutation that is **probably** responsible for a modest cardiomyopathy in the C57BL/6N substrain

The authors need to include the significance values for the echo data in Suppl Table I. **These have been added in the new Excel file for Supplementary table 1. Significance data are displayed for each pairwise comparison within the 3 datasets.**

To make it easier for the reader to follow the study and interpret the data, the authors should use more descriptive terms.

To ascertain whether the B6J substrain developed cardiomyopathy, male mice from the three substrains B6N, B6J and B6J-BAC ??? were analysed at 3, 12 and 18 months of age, **All mentions of B6J-BAC have been changed to B6J-Nnt, in both the text and figures.**

The authors should refer to the B6J rescue mouse as B6J-Nnt instead of B6J-BAC throughout the paper. **We thank the reviewer for this suggested change, all mentions of B6J-BAC have been changed to B6J-Nnt, in both the text and figures.**

241 Luciferase activity was significantly lower in HEK293T cells transfected with the 'B6N-Luc' ??? vector compared to those transfected with 'B6J-Luc', indicating the mutation is sufficient to abrogate protein translation (Fig 6b). **This is outlined in the methods, and we have also changed the nomenclature of these vectors from B6N-Luc to B6N-5'UTR-Mylk3-Luc and B6J-Luc to B6J-5'UTR'-Mylk3-Luc.**

To help the reader, the authors should specify the luciferase constructs, i.e., refer to B6N-Luc as B6N-5UTR-MYLK3 and B6J-Luc as B6J-5UTR-MYLK3. Or simply B6J-5UTR. **This is outlined in the methods, and we have also changed the nomenclature of these vectors from B6N-Luc to B6N-5'UTR-Mylk3-Luc and B6J-Luc to B6J-5'UTR'-Mylk3-Luc.**

We believe these changes satisfactorily address the points raised here and render the manuscript ready for publication.

March 10, 2020

RE: Life Science Alliance Manuscript #LSA-2019-00593-TRRR

Prof. Louise Metherell
Queen Mary University Of London
Centre for Endocrinology,
Barts and the London School of Medicine and Dentistry, Queen Mary University of London
London EC1M 6BQ
United Kingdom

Dear Dr. Metherell,

Thank you for submitting your Research Article entitled "Myk3 null C57BL/6N mice develop cardiomyopathy whereas Nnt null C57BL/6J do not". It is a pleasure to let you know that your manuscript is now accepted for publication in Life Science Alliance. Congratulations on this interesting work.

DISTRIBUTION OF MATERIALS:

Again, congratulations on a very nice paper. I hope you found the review process to be constructive and are pleased with how the manuscript was handled editorially. We look forward to future exciting submissions from your lab.

Sincerely,

Andrea Leibfried, PhD
Executive Editor
Life Science Alliance
Meyerohofstr. 1
69117 Heidelberg, Germany
t +49 6221 8891 502
e a.leibfried@life-science-alliance.org
www.life-science-alliance.org